# Distinct roles for two *Caenorhabditis elegans* acid-sensing ion channels in an ultradian clock

Eva Kaulich[1], Trae Carroll[2], Brian D Ackley[3], Yi-Quan Tang[1,4], Iris Hardege[1], Keith Nehrke[5], William R Schafer[1,6]*, Denise S Walker[1]*

[1]Neurobiology Division, MRC Laboratory of Molecular Biology, Cambridge, United Kingdom; [2]Department of Pathology and Lab Medicine, University of Rochester Medical Center, Rochester, United States; [3]Department of Molecular Biosciences, University of Kansas, Lawrence, United States; [4]State Key Laboratory of Medical Neurobiology and MOE Frontiers Center for Brain Science, Institutes of Brain Science, Fudan University, Shanghai, China; [5]Department of Medicine, Nephrology Division, University of Rochester Medical Center, Rochester, United States; [6]Department of Biology, KU Leuven, Leuven, Belgium

*For correspondence:
wschafer@mrc-lmb.cam.ac.uk (WRS);
dwalker@mrc-lmb.cam.ac.uk (DSW)

**Competing interest:** The authors declare that no competing interests exist.

**Abstract** Biological clocks are fundamental to an organism's health, controlling periodicity of behaviour and metabolism. Here, we identify two acid-sensing ion channels, with very different proton sensing properties, and describe their role in an ultradian clock, the defecation motor program (DMP) of the nematode *Caenorhabditis elegans*. An ACD-5-containing channel, on the apical membrane of the intestinal epithelium, is essential for maintenance of luminal acidity, and thus the rhythmic oscillations in lumen pH. In contrast, the second channel, composed of FLR-1, ACD-3 and/or DEL-5, located on the basolateral membrane, controls the intracellular $Ca^{2+}$ wave and forms a core component of the master oscillator that controls the timing and rhythmicity of the DMP. *flr-1* and *acd-3/del-5* mutants show severe developmental and metabolic defects. We thus directly link the proton-sensing properties of these channels to their physiological roles in pH regulation and $Ca^{2+}$ signalling, the generation of an ultradian oscillator, and its metabolic consequences.

## Editor's evaluation

In this study, Kaulich and colleagues report an intriguing interplay of different proton-sensitive ion channels at different locations in the intestine of *C. elegans*. They provide both in-depth biophysical characterisations of different channel complexes as well as their in vivo involvement in the generation of calcium waves for the rhythmic defecation motor program of the worm.

## Introduction

Many physiological processes occur with a predictable periodicity. The maintenance of such rhythms can rely on fluctuating gene expression, hormone concentrations, or homeostatic oscillations in signalling molecules within cellular compartments, depending on the timescale of the clock (*Hastings et al., 2018*; *Rensing et al., 2001*; *Branicky and Hekimi, 2006*; *Thomas, 1990*). Disruption of biological clocks can have devastating consequences on homeostasis or behaviour (*Marcheva et al., 2013*). Here, we examine the physiological role of acid-sensing ion channels (ASICs) in the defecation motor program (DMP) of the nematode *Caenorhabditis elegans*, an ultradian rhythm that occurs about every

**eLife digest** Biological clocks regulate a myriad of processes that occur periodically, from sleeping and waking to how cells use nutrients and energy. One such clock is the one that controls intestinal movements and defecation in the nematode worm *Caenorhabditis elegans*, which consists of three muscle contractions occurring every 50 seconds. This rhythm is controlled by calcium and proton signalling in the cells of the intestine.

The cells of the nematode intestine form a tube, through which gut contents pass. The inside of the tube is acidic, but acidity also plays a role on the outer face of the intestinal tube. In this area, nutrients are distributed and signals are conveyed to other tissues, such as muscles. In fact, acid – in the form of protons – secreted from the intestinal cells stimulates the muscles that contract in the biological clock that controls the worms' defecation. However, it is poorly understood how the worms control the release of these protons.

Kaulich et al. identified two ion channels on the membranes of intestinal cells that become inhibited when the levels of acid surrounding them are high. These channels play distinct roles in controlling the contractions that move the contents of the roundworms' intestines along. The first channel contains a protein called ACD-5, and it is in the membrane of the intestinal cells that faces the inside of the intestinal tube. The second channel is formed by three proteins: FLR-1, ACD-3 and DEL-5. This channel is found on the other side of the intestinal cells, the region where nutrients are distributed and signals are conveyed to the rest of the body.

To determine the role of each channel, Kaulich et al. genetically engineered the worms so they would not make the proteins that make up the channels, and imaged the live nematodes to see the effects of removing each channel. The inside of the intestines of worms lacking the ACD-5 containing channel was less acidic than that of normal worms, and the timing of the contractions that control defecation was also slightly altered. Removing the second channel (the one formed by three different proteins), however, had more dramatic effects: the worms were thin, developed more slowly, had less fat tissue and defecated very irregularly.

Kaulich et al. imaged live worms to show that the second channel plays a major role in regulating oscillations in acidity both inside and outside cells, as well as controlling calcium levels. This demonstrates that this channel is responsible for the rhythmicity in the contractions that control defecation in the nematodes. Their findings provide important insights towards better understanding proton signalling and the role of acid-sensing ion channels in cellular contexts and biological clocks.

50 s (*Thomas, 1990*; *Branicky and Hekimi, 2006*). We take advantage of this short-period cellular oscillator to relate ion channel physiology *in vivo* to organismal metabolism and behaviour.

ASICs, the acid-sensing members of the Degenerin/Epithelial Sodium Channel (DEG/ENaC) superfamily of cation channels, are thought to be the main proton receptors in vertebrates (*Soto et al., 2018*; *Du et al., 2017*). They are an important drug target because acidosis is a feature of painful inflammatory and ischemic conditions. ASICs are widely expressed in the central and peripheral nervous system, where their roles in inflammation, ischemia, pain perception, and learning are extensively studied (*Waldmann et al., 1997*; *Wemmie et al., 2002*; *Wemmie et al., 2003*; *Voglis and Tavernarakis, 2008*; *Kreple et al., 2014*; *Du et al., 2014*). However, less is known about their physiological role in non-neuronal tissue such as the intestinal epithelium, despite cross-phyla evidence for gastrointestinal expression and the role of acidity therein (*Dong et al., 2011*; *Schaefer et al., 2000*; *Take-uchi et al., 1998*; *Levanti et al., 2011*). The extreme acidity of the stomach requires an elaborate system of mucosal protective mechanisms, failure of which results in conditions such as gastritis, ulceration, and dyspepsia. Acidification of the intestinal tract is also a hallmark of conditions such as irritable bowel syndrome and cystic fibrosis, so understanding the role of acid sensors, of which the ASICs are prime candidates, is of wide-ranging therapeutic importance (*Holzer, 2007*).

The DMP depends on inositol 1,4,5-trisphosphate ($IP_3$)-dependent $Ca^{2+}$ oscillations in the intestinal epithelial cells. $IP_3$ signalling loss-of-function mutations cause dramatic increases in both period length and variability, whereas overexpression of the $IP_3$ receptor, ITR-1, reduces period length (*Dal Santo et al., 1999*; *Walker et al., 2002*; *Espelt et al., 2005*). Increased cytosolic $Ca^{2+}$ results in the rhythmic release of protons from the basolateral membrane, into the pseudocoelom. This triggers the posterior

body wall muscles to contract (pBoc), via activation of a proton-gated cation channel PBO-5/PBO-6 (*Beg et al., 2008*; *Pfeiffer et al., 2008*). Propagation of the $Ca^{2+}$ wave from the posterior to anterior then initiates contraction of the body wall muscle at the anterior of the intestine (aBoc) a few seconds later, via activation of the motor neuron AVL, and then finally, GABA release from both AVL and DVB triggers the enteric muscle contraction (EMC), expelling gut contents (*Beg and Jorgensen, 2003*; *McIntire et al., 1993*). Premature termination of the $Ca^{2+}$ wave or pharmacologically blocking the $Ca^{2+}$ wave propagation before it can reach the anterior disrupts aBoc and EMC (*Teramoto and Iwasaki, 2006*), further supporting the idea that $Ca^{2+}$ is the master regulator.

In addition to $Ca^{2+}$ and pseudocoelomic pH, proton oscillations in the intestinal lumen are also integral to the DMP. The luminal pH fluctuates between ~4.4 and 6.5 during the cycle and the pH transitions in a wave that travels from posterior to anterior (*Beg et al., 2008*; *Allman et al., 2009*; *Pfeiffer et al., 2008*; *Bender et al., 2013*; *Chauhan et al., 2013*), dependent on proton uptake and release at the apical membrane. Proton gradients are fundamental to nutrient absorption, and some DMP-affecting mutations can result in aberrant fat metabolism and deficient nutrient uptake (*Sheng et al., 2015*; *Allman et al., 2009*). Loss of proton regulation can also disrupt immunity and pathogen resistance, as seen in mutants for *C. elegans* CHP1 homolog, PBO-1, in which the lumen is constantly acidic (*Benomar et al., 2020*). Thus, dysregulation of components of this ultradian clock can have far reaching organismal consequences.

The precise relationship between luminal proton oscillations, intestinal $Ca^{2+}$ and the DMP is unclear; but the mutant phenotypes of some of the known transporters and exchangers that control lumen pH provide some clues. Knockdown of VHA-6, a subunit of the apical proton-transporting V-type ATPase, leads to more neutral lumen pH and an increase in cycle length (*Allman et al., 2009*). For mutations in the $Na^+/H^+$ exchanger PBO-4, the effect on lumen pH is controversial, since measurement using KR35 dye (*Bender et al., 2013*; *Benomar et al., 2020*) reveals a defect in anterior acidification, whereas measurement using Oregon Green 488 (*Pfeiffer et al., 2008*) reveals that global pH oscillations are normal; cycle length and $Ca^{2+}$ oscillations are unaffected (*Thomas, 1990*; *Pfeiffer et al., 2008*; *Benomar et al., 2020*; *Bender et al., 2013*; *Beg et al., 2008*). Mutants of *pbo-1*, on the other hand, have a highly acidic lumen and no proton wave, but $Ca^{2+}$ oscillations and cycle length are largely unaffected (*Wagner et al., 2011*; *Thomas, 1990*; *Pfeiffer et al., 2008*; *Benomar et al., 2020*). So, there is some evidence that the periodicity of the DMP can be independent of luminal pH, and thus that proton dynamics would be downstream of the $Ca^{2+}$ oscillator.

We and others hypothesized that a proton wave transitioning from the posterior to the anterior of the lumen could lead to sequential activation of proton sensors in the apical membrane (*Chauhan et al., 2013*; *Bender et al., 2013*; *Benomar et al., 2020*), and acid-sensing DEG/ENaCs are good candidates. Compared to mammals, the *C. elegans* genome encodes a vastly expanded array of 30 DEG/ENaCs. One of these, FLR-1 (FLuoride-Resistant phenotype), functions in DMP rhythmicity and execution (*Kwan et al., 2008*; *Take-uchi et al., 1998*), although, the mechanistic basis of this role is unclear. With the exception of ACD-1, the first acid-inactivated DEG/ENaC to be identified (*Wang et al., 2008*), the proton sensing properties of the *C. elegans* DEG/ENaCs remained uncharacterised.

Here, we address the physiological role of *C. elegans* acid-sensing DEG/ENaCs in the intestine. We explore how $Ca^{2+}$ and proton oscillations maintain periodicity and execution of the DMP, and find that disruption of $Ca^{2+}$ and proton homeostasis are detrimental to metabolism and organismal health. We show that four intestinal DEG/ENaC subunits form two acid-inactivated channels with distinct properties. The first contains ACD-5 and localises to the luminal face of the intestinal cells and acts as a proton sensor, contributing to the establishment of low luminal pH and thus enabling cyclic pH oscillations. The second channel, containing FLR-1, ACD-3, and/or DEL-5, localises to the basolateral membrane, facing the pseudocoelom, and is essential for the $Ca^{2+}$ oscillations that control the timing and execution of the DMP and thus is a key component of this ultradian clock. We thus provide direct evidence linking the channel properties of these proton sensors to their roles in pH regulation and $Ca^{2+}$ signalling *in vivo*, which can provide insight into potential cellular roles for other acid-sensing members of this important channel family.

## Results

### ACD-5 forms a homomeric proton-sensing channel localised to the apical membrane

Acid-sensing members of the DEG/ENaC family can form homomeric or heteromeric channels that are activated and/or inhibited by low pH (*Zhang and Canessa, 2002*; *Wang et al., 2008*; *Elkhatib et al., 2019*). Many, but not all, are blocked by the anti-hypertensive amiloride; indeed, in some cases amiloride can potentiate currents (*Canessa et al., 1994*; *Fechner et al., 2021*; *Besson et al., 2017*). We expressed all 30 *C. elegans* DEG/ENaC subunits in *Xenopus* oocytes and assayed for activity using Two-electrode-voltage clamp (TEVC) recording, identifying three channels that were activated by low pH and five that were inhibited (*Kaulich et al., 2022*). Oocytes expressing the ACD-5 subunit showed currents that could be inhibited by amiloride (*Figure 1A and B*), and showed robust responses to pH. ACD-5 exhibited maximal current at around pH 6 and was inhibited by both higher and lower pH, with $pH_{50}$s of 4.87 and 6.48 (*Figure 1C*).

We next investigated ion selectivity, assessing the shift in reversal potential ($\Delta E_{rev}$) by substituting NaCl with either equimolar KCl, LiCl, or a $CaCl_2$ (*Hardege et al., 2015*; *Wang et al., 2008*). Monovalent ion-substitution experiments were conducted in the absence of $Ca^{2+}$, as $Ca^{2+}$ can block some vertebrate ASICs (*Paukert et al., 2004*). Permeability for $Ca^{2+}$ (*Wang et al., 2008*) was also tested, as some constitutively open *C. elegans* DEG/ENaCs are permeable for $Ca^{2+}$ in addition to monovalent cations (*Matthewman et al., 2016*). ACD-5 shows a preference for $Li^+$ over $Na^+$ and a small preference of $K^+$ over $Na^+$ and is $Ca^{2+}$ impermeable (*Figure 1D*). ACD-5 was not permeable for protons, as the $\Delta E_{rev}$ did not show a positive shift when extracellular proton concentration was increased, and removal of $Ca^{2+}$ from the solution also did not alter $\Delta E_{rev}$. By contrast, removal of $Na^+$ (by substituting with NMDG) induced a large negative $\Delta E_{rev}$ (*Figure 1—figure supplement 1A-D*). Thus, ACD-5 forms a homomeric amiloride-sensitive, pH-sensitive monovalent cation channel.

To determine the function of the ACD-5 channel, we investigated its mutant phenotype. The only *acd-5* mutation available was *acd-5(ok2657),* generated by the *C. elegans* Deletion Mutant Consortium (*C. elegans Deletion Mutant Consortium, 2012*), a truncation lacking TM2 and the C-terminus (*Figure 1—figure supplement 2A,B*). To determine the effect of this mutation on channel function, we expressed the mutant form in *Xenopus* oocytes. Oocytes expressing ACD-5(ok2657) alone did not display any currents and were unaffected by amiloride (*Figure 1—figure supplement 3A,B*). However, co-expressing ACD-5(ok2657) with wild-type ACD-5 significantly altered the expressed currents. Specifically, increasing the ratio of injected *acd-5(ok2657)* cRNA to wild-type *acd-5* cRNA resulted in a decrease in current (*Figure 1E*). This suggests that the ACD-5(ok2657) subunit either forms a non-functional heteromer or sequesters factors required for wild-type ACD-5 function, and thus is a dominant mutation, hindering functional channel formation.

Given ACD-5's dual response to protons, we investigated its subcellular localisation to understand its physiological pH environment. A translational *acd-5::mKate2* C-terminal fusion showed expression in the intestinal epithelial cells, localising to the apical membrane, as confirmed by co-localisation with an apical membrane marker (OPT-2::GFP [*Nehrke, 2003*]) and by comparing the localisation with that of markers for the intestinal cytoplasm (*Pdel-5::GFP*) and lumen (ingested fluorescent dye, DiO) (*Figure 1F and G* and *Figure 1—figure supplement 4*). The observation that ACD-5 localises to the apical membrane indicated that its physiological environment is likely in the range pH 4–6, the pH range of the intestinal lumen (*Pfeiffer et al., 2008*; *Chauhan et al., 2013*), consistent with our electrophysiological findings that ACD-5 exhibits maximal current at pH 6 and is inhibited at pH 4.

### Mutations in acd-5 dramatically alter intestinal lumen pH

Given its luminal location and physiological sensitivity to mildly acidic pH, we hypothesised ACD-5 may have an impact on the acidity of the lumen. To test this, we used the ingested acid-activated fluorophore Kansas Red (KR35). KR35 is fluorescent when protonated and has a pKa of 3.5 (where 50% of the molecule is fluorescent at pH=3.5). At lower pH ranges, the molecule is fluorescent and it is non-fluorescent in environments of pH>5.5. The rapid conversion between fluorescent and non-fluorescent forms allows visualisation of the dynamic wave in luminal pH. Every 50 s, the posterior fluorescence transitions to the anterior region of the intestine, remaining there for 3–7 s, a period referred to as the maximum anterior transition (MAT) (*Bender et al., 2013*; *Benomar et al., 2020*). We analysed the dominant *acd-5(ok2657)* as well as a null mutant generated using CRISPR/Cas9 (*Dokshin*

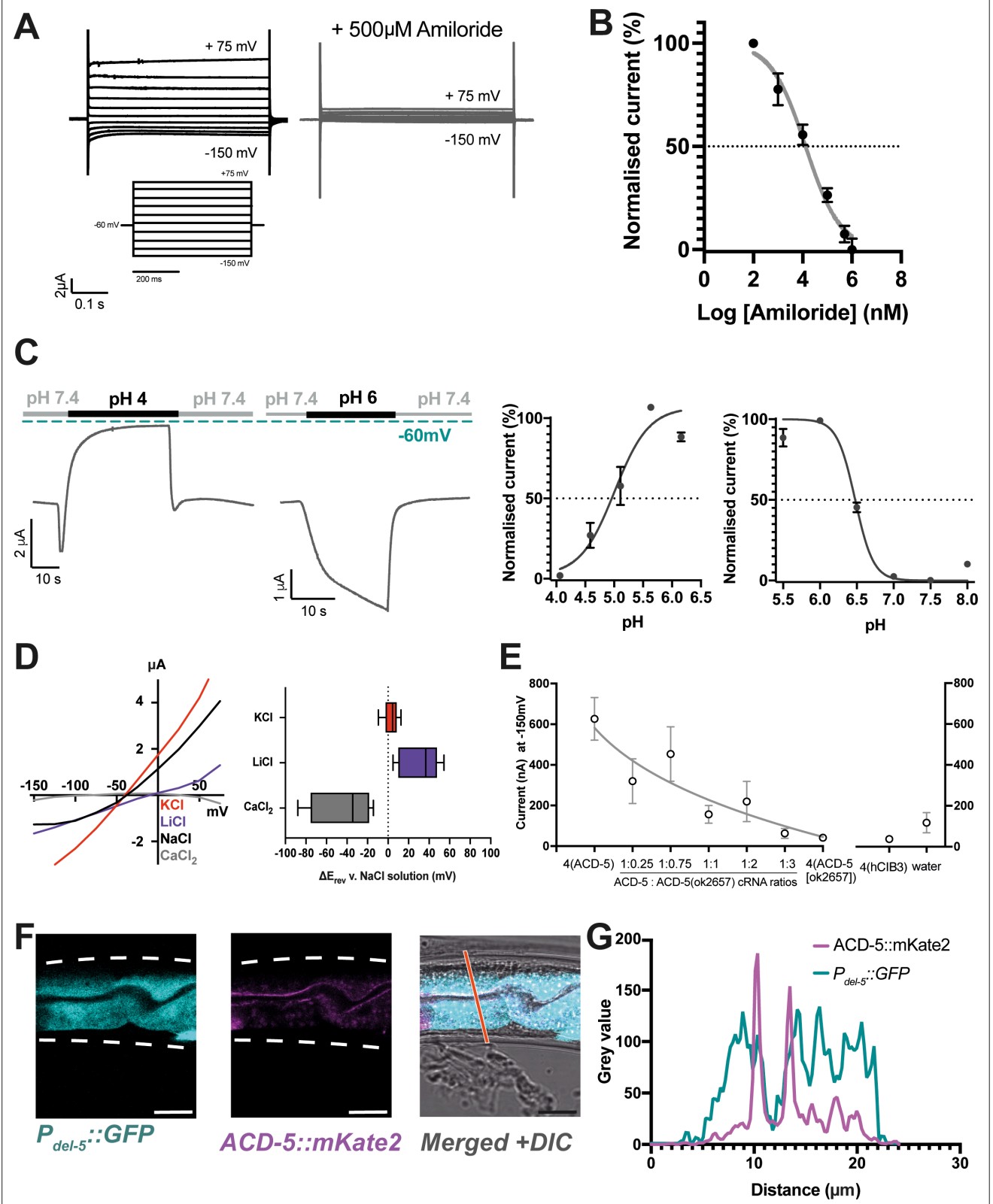

**Figure 1.** ACD-5 is an amiloride-sensitive, acid-sensing cation channel on the luminal membrane of the intestine. (**A**) ACD-5 currents are amiloride-sensitive. Representative ACD-5 transients in *Xenopus* oocytes in the presence and absence of amiloride and (**B**) normalised amiloride-dose response curve ($I/I_{max}$ as percentage) for ACD-5, showing a half-maximal inhibitory concentration ($IC_{50}$) of 131μM ($LogIC_{50}$=4.118) (N=8) indicated by the dashed line (curve fitted as described below). Error bars represent mean ± SEM. (**C**) ACD-5 can form a homomeric acid-sensing ion channel. Left panel:

*Figure 1 continued on next page*

Kaulich *et al*. eLife 2022;11:e75837. DOI: https://doi.org/10.7554/eLife.75837                                      5 of 37

*Figure 1 continued*

Representative trace of ACD-5-expressing *Xenopus* oocytes subjected to a down-step to pH 4 or 6. Currents were recorded at a holding potential of –60mV and traces are baseline-subtracted and drift-corrected using the Roboocyte2+ (Multichannels) software. Right panel: pH-dose dependence (I/I$_{max}$ expressed as percentage). ACD-5-expressing oocytes were perfused with solutions of decreasing pH from pH 6 (left), showing an inhibitory pH$_{50}$ of 4.90 (N=5); when perfused with increasing pH from pH 6 (right), showing an excitatory pH$_{50}$ of 6.48 (N=7). Currents were recorded at a holding potential of –60mV, normalised to maximal currents and best fitted with the Hill's equation (Nonlin fit Log(inhibitor) versus normalised response – variable slope) in GraphPad Prism. Error bars represent mean ± SEM. (**D**) Current-voltage (IV) curve (left) and change in reversal potential, ΔE$_{rev}$ when shifting from a NaCl solution to CaCl$_2$, KCl, or LiCl (right). N=12 oocytes. Presented as median and IQR (Tukey method). (**E**) ACD-5(ok2657) is a dominant mutation. Expression of varying ratios of ACD-5(ok2657) with a constant concentration of wild-type ACD-5 in *Xenopus* oocytes. hCIB3 (Calcium and integrin-binding family member 3) was used as a filler to account for amount of cRNA injected. Mean ± SEM. 9<N > 19 for each ratio. (**F**) ACD-5 is localised at the apical (luminal) intestinal membrane. Localisation of mKate2-C-terminally tagged ACD-5 (*ljEx1470 (Pacd-5::acd-5(no stop) cDNA::mKate2)*) shows apical membrane localisation (magenta). Cytoplasm of the intestinal cells is shown in cyan (*ljEx1349 (Pdel-5::GFP)*). (**G**) Intensity profile taken at the part of the intestine indicated by the red line. Scale bar: 10μm.

The online version of this article includes the following figure supplement(s) for figure 1:

**Figure supplement 1.** Further characterisation of ACD-5 homomeric channel properties.

**Figure supplement 2.** Genomic regions and schematic of predicted protein structures of the intestinal DEG/ENaCs, showing mutations used in this study.

**Figure supplement 3.** Characterisation of ACD-5 channels carrying the *ok2657* mutation in *Xenopus* oocytes.

**Figure supplement 4.** ACD-5 localises to the apical membrane.

*et al., 2018*), *acd-5(lj122)* (*Figure 1—figure supplement 2A,B*). Mutants carrying either *acd-5* allele showed overall lower fluorescence intensities, indicating a more neutral lumen pH compared to wild-type (*Figure 2A and B*). In the *acd-5* mutants, the dye in the posterior part of the intestine was too faint to quantify (in the non-fluorescent range of KR35 (pH 5.5–3.5)), this suggests that the increased luminal pH is a phenomenon that affects the whole intestinal lumen. This is in line with the expression data that ACD-5 localises to the luminal membrane throughout the intestine. As a result, both *acd-5(lj122)* and *acd-5(ok2657)* mutants exhibited a high incidence of missed (or undetectable) MATs, with only 61% and 56 %, respectively, displaying at least one detectable MAT in the 2-min recording time, compared with 82% of wild-type animals (*Figure 2C*). These defects were rescued in the *acd-5(lj122)* null by expression of the wild-type *acd-5* gene, but rescue was not observed for the truncated *acd-5(ok2657)* (*Figure 2A, B and C*). Where detectable, MAT interval length was not grossly affected (*Figure 2D*). Thus, although ACD-5 apparently plays a pivotal role in the control of proton concentration, it does not greatly influence the timing of the luminal proton wave.

## Mutations in *acd-5* subtly alter intestinal Ca²⁺ oscillations and DMP behaviour

Mutations that disrupt proton homeostasis in the intestinal lumen can have differing effects on intestinal Ca$^{2+}$ transients and DMP behavioural output. Therefore, we assessed whether *acd-5* affects intestinal Ca$^{2+}$ oscillations, the presumed master oscillator for the DMP. Both alleles displayed intact, regular Ca$^{2+}$ oscillations, with small changes in timing and amplitude. *acd-5(ok2657)* mutants showed an increased amplitude and significantly longer intervals between Ca$^{2+}$ peaks, whereas *acd-5(lj122)* null animals displayed a reduced amplitude and a very small decrease in interval length (*Figure 3A and B* and *Figure 3—figure supplement 1* for representative example traces). This shows that *acd-5* does interact with the Ca$^{2+}$ oscillations, albeit in a relatively subtle manner.

To explore whether the changes observed in Ca$^{2+}$ oscillation translate into behavioural outputs, we assessed the timing and execution of DMP cycles. The cycle intervals observed correspond well with the Ca$^{2+}$ imaging observations; *acd-5(ok2657)* exhibited significantly longer intervals and *acd-5(lj122)* animals showed no difference from wild-type (*Figure 3C*). The distinct phenotypes observed for these two alleles were consistent with our *in vitro* evidence that *acd-5(ok2657)* is a dominant mutation. Heterozygous animals and wild-type animals overexpressing the *acd-5(ok2657)* mutant gene, under the endogenous promoter or under another intestinal promoter (*Pges-1*) all retained the longer cycle phenotype, supporting this hypothesis (*Figure 3C*). We also found that overexpression of the *acd-5(ok2657)* gene subtly increased variability, although since this was also true of overexpression of the wild-type *acd-5* gene (*Figure 3—figure supplement 2A,B*), it may have been a consequence of overexpression rather than the *acd-5(ok2657)* mutation itself.

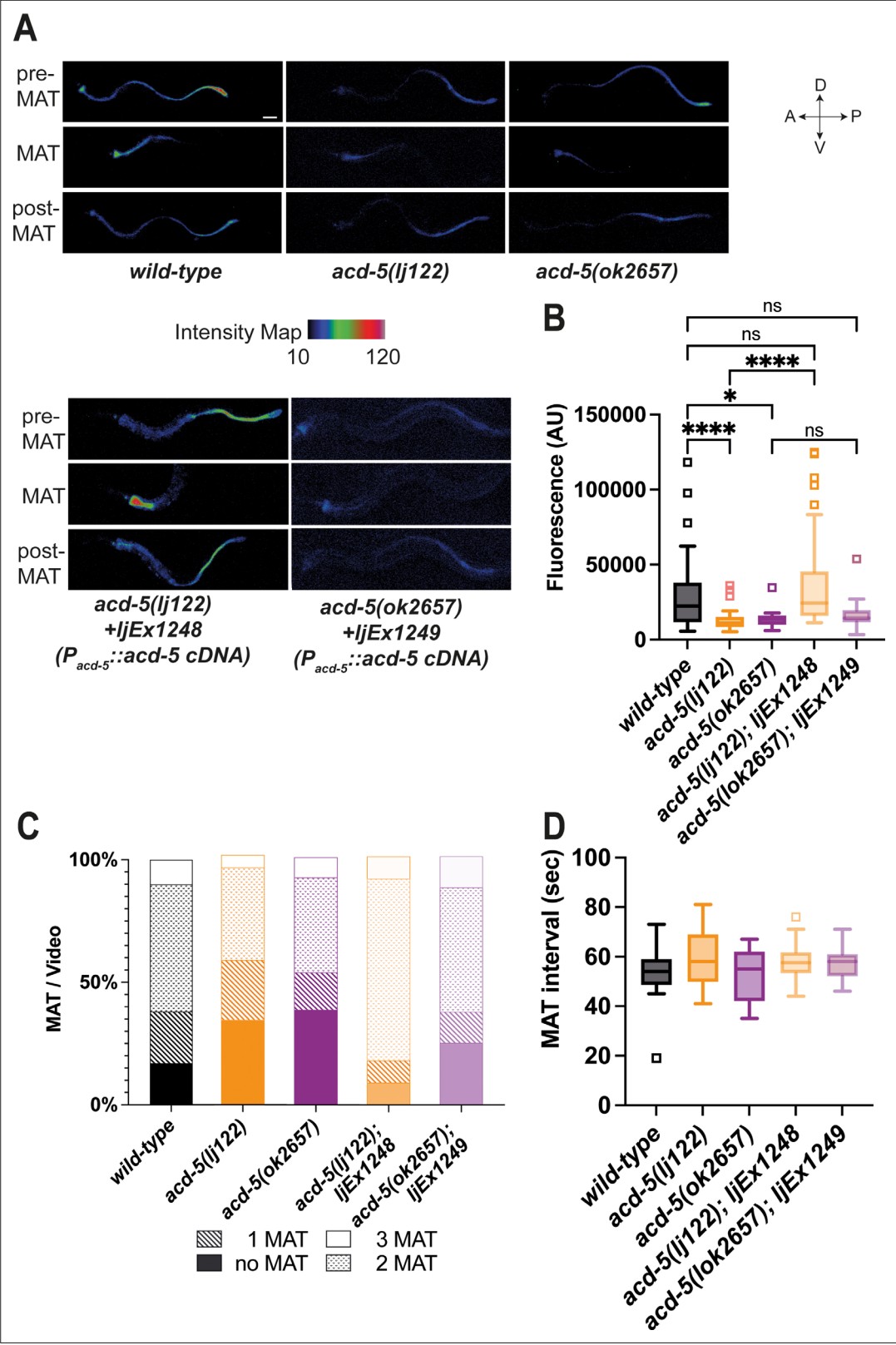

**Figure 2.** ACD-5 mutations show disruptions in intestinal lumen acidity and luminal proton oscillations. (**A**) ACD-5 is important to establish and maintain an acidic intestinal pH. Heat map of pixel fluorescence intensity with red representing highest intensity (high acidity), and black lowest intensity (low acidity), after feeding on the pH-sensitive probe Kansas Red (KR35) for 30min (10µM). Maximum anterior transition (MAT)±15s (pre/post-MAT)

*Figure 2 continued on next page*

*Figure 2 continued*

during the DMP extracted from videos of free-moving animals are shown. Rescue lines express the *ljEx1248(Pacd-5::acd-5 cDNA)* or *ljEx1249(Pacd-5::acd-5 cDNA)* extrachromosomal array, respectively. D=dorsal, V=ventral, A=anterior, P=posterior. Scale bar: 50µm. (**B**) *acd-5(lj122)* mutants (****p<0.0001) and *acd-5(ok2657)* mutants (*p=0.0309) display lower intestinal lumen pH compared to the wild-type. This could be rescued only in the *acd-5(lj122)* mutants (****p<0.0001). Quantification of fluorescence in the MAT. Data are presented as median and IQR (Tukey method, N=24, 19, 9, 20, and 12, respectively). Outliers are indicated by squared boxes. Kruskal-Wallis test with Dunn's multiple comparison test. (**C**) *acd-5* mutants display a reduced number of MATs. Quantification of MATs per 2-min video recording shown as percentages of animals displaying 1, 2 or 3 MATs. Median and IQR (Tukey method, N=29, 38, 13, 23, and 16, respectively). (**D**) MAT intervals are not affected by *acd-5* mutations. Time interval of MAT of *acd-5* mutants, wild-type and rescues. Kruskal-Wallis test with Dunn's multiple comparison test was non-significant (P=0.405). Data are presented as median and IQR (Tukey method). N=21, 18, 7, 20, and 12, respectively.

To gain further insight into the role of *acd-5* in the DMP, we examined the genetic interaction between *acd-5* alleles and known DMP factors. The IP$_3$ receptor, ITR-1, plays a critical role in the Ca$^{2+}$ oscillations that drive the DMP and the loss-of-function allele *itr-1(sa73)* results in a dramatic increase in both cycle length and variability (*Dal Santo et al., 1999*; *Walker et al., 2002*; *Espelt et al., 2005*). We observed the previously described compensatory effect (*Kwan et al., 2008*) in *itr-1(sa73)*, where an aberrantly long cycle is followed by a short one, and vice versa (*Figure 3—figure supplement 3A*). Although the *acd-5* alleles did not significantly reduce the cycle length or variability of *itr-1(sa73)*, and the compensatory effect remained intact, we did nevertheless observe a striking suppression of excessively long cycles, with an almost complete loss of cycles over 200 s (*Figure 3D and E* and *Figure 3—figure supplement 3A-C*). A Kolmogorov-Smirnov comparison of the cumulative distribution showed that cycle length distribution was not significantly different, but the CV distribution was significantly different for both *itr-1/acd-5* double mutants (*Figure 3—figure supplement 4A,B*). This suggests that *acd-5* is responsible for the dysregulation that results in those longer, more variable cycles, perhaps suggesting a role for ACD-5 in regulating cycle length that is otherwise presumably masked or overridden by the dominant influence of the Ca$^{2+}$ oscillator.

The Na$^+$/H$^+$ exchanger PBO-4 localises to the basolateral membrane of the posterior intestinal cells, where it plays a central role in the release of protons into the pseudocoelom to trigger pBoc, and mutations result in missed or weak pBoc and frequently missed EMCs (*Benomar et al., 2020*; *Beg et al., 2008*; *Pfeiffer et al., 2008*), but the precise relationship between Ca$^{2+}$ increase and PBO-4-mediated proton oscillations is unknown. PBO-4 also localises to the apical membrane (*Bender et al., 2013*) but, as discussed earlier, its role there is controversial. In view of a potential apical role, we investigated its interaction with *acd-5*. The *pbo-4* mutation was dominant over the *acd-5* mutations, with *acd-5/pbo-4* double mutants both displaying wild-type cycle length and variability (like the *pbo-4* single), and EMC and pBoc defects similar to the *pbo-4* mutant (*Figure 3F*, *Figure 3—figure supplement 3D-F*). This is consistent with *pbo-4* acting genetically upstream of *acd-5*; however, this could reflect an apical role for PBO-4 (*Figure 3G*) or simply the dominance of basolateral proton release in driving downstream events.

## FLR-1, ACD-3, and DEL-5 localize to the basolateral membrane and can form acid-sensitive heteromeric channels

Previous research identified the expression of another DEG/ENaC channel gene, *flr-1*, in the intestine (*Take-uchi et al., 1998*), and it was shown to localise to both apical and basolateral membranes (*Take-uchi et al., 1998*; *Kobayashi et al., 2011*). *flr-1* mutants show more severe disruption of the DMP than *acd-5* mutants, leading us to hypothesise that there might be two distinct channels in the intestinal cell: A heteromeric or homomeric ACD-5-containing channel on the apical membrane and an FLR-1-containing channel on both membranes. Since FLR-1 expressed alone in oocytes did not produce pH-sensitive currents in our experiments, we hypothesised that FLR-1 could contribute to a heteromeric channel. Therefore, we used transcriptional fusions to search for additional intestinally expressed DEG/ENaCs. We found two such intestine-expressed genes, *acd-3* and *del-5* (*Figure 4—figure supplement 1*). Whereas ACD-5 and FLR-1 share the highest homology with mammalian ASIC2, ACD-3 shares the highest homology with mammalian ASIC5 and DEL-5 is more distantly

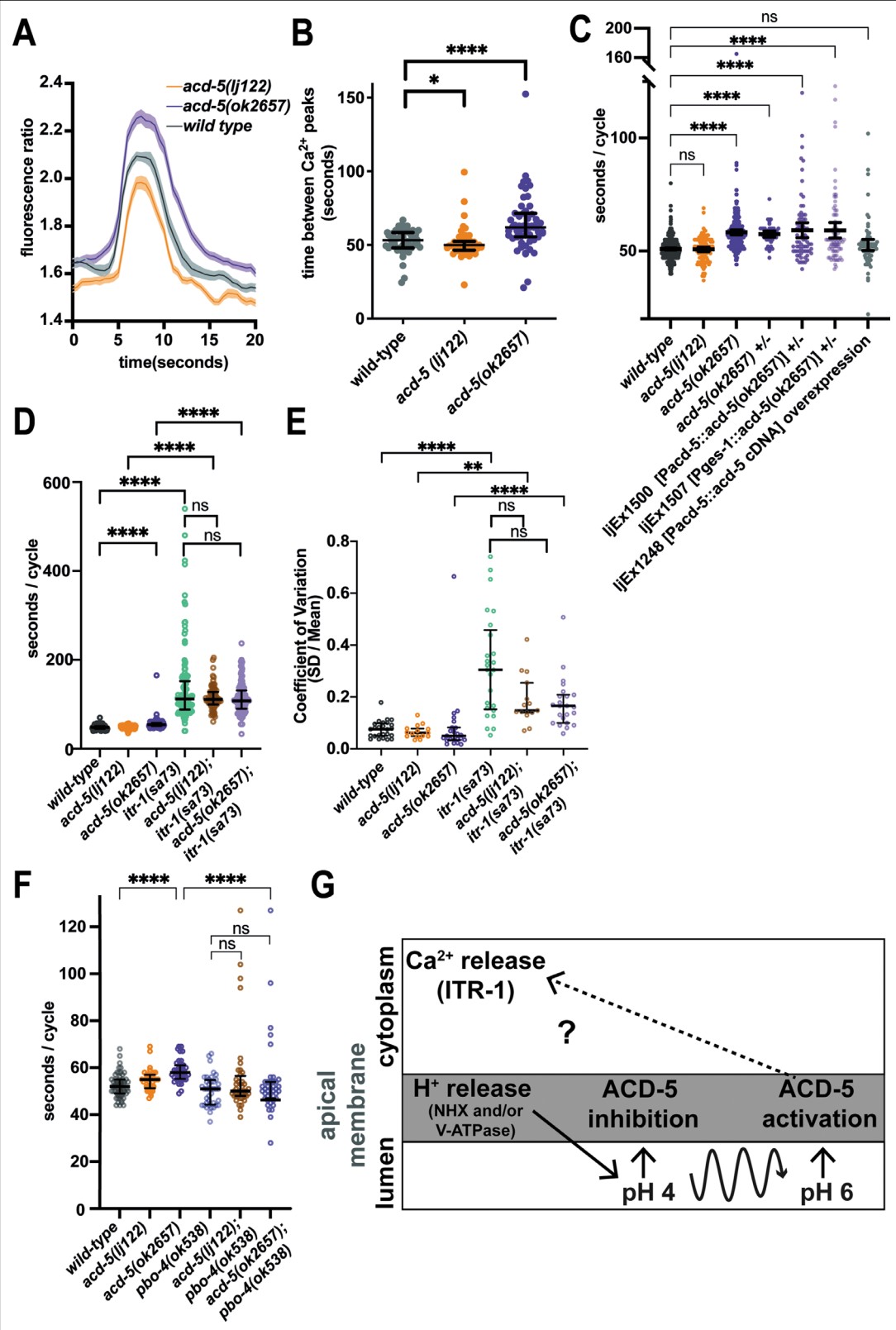

**Figure 3.** ACD-5 mutations affect Ca²⁺ oscillations and behaviour. (**A**) and (**B**) Ca²⁺ imaging of the intestinal cells of freely moving, defecating animals. (**A**) *acd-5* mutations affect intestinal Ca²⁺ oscillations. Ca²⁺ mean trace± SEM of fluorescence ratio for *acd-5* mutants and wild-type. Ca²⁺ 'peaks' were centrally aligned as described in Materials and methods. (**B**) Ca²⁺ cycle length. Mann-Whitney U between wild-type and *acd-5(ok2657)* ****p<0.0001 (indicating a longer cycle interval), and between wild-type and *acd-5(lj122)* *p=0.0266 (indicating a shorter cycle interval). Error bars represent median

*Figure 3 continued on next page*

*Figure 3 continued*

and IQR, N=50, 55, and 53, respectively (i.e., individual intervals from 15 animals for each genotype, recorded for 5min each). (**C**) ACD-5(ok2657) is a dominant mutation. Comparison of defecation cycle length in *acd-5* mutant and overexpression strains. Kruskal-Wallis test and post hoc Dunnett's multiple comparisons test found no significant difference between wild-type and *acd-5(lj122)* animals (p>0.9999) but a significant difference from wild-type for homo- and heterozygous *acd-5(ok2657)* animals and wild-type animals expressing *ljEx1500(Pacd-5::acd-5[ok2657])* or *ljEx1507(Pges-1::acd-5[ok2657])* (****p<0.0001); those overexpressing wild-type *ljEx1248(Pacd-5::acd-5cDNA)* were not significantly different from wild-type (p>0.9999). Median and IQR. 50<N<75 (10–15 animals for each group, five cycles for each animal). (**D**) and (**E**) Effect of *acd-5* alleles on the defecation interval of *itr-1(sa73)* animals. (**D**) DMP interval length (median and IQR). Kruskal-Wallis test was significant (****p<0.0001). A post hoc Dunn's multiple comparison test indicated that compared to the wild-type, *acd-5(ok2657)* mutants and the *itr-1* single and double mutants (****p<0.0001) all showed a significant increase in interval length. (**E**) Coefficient of variation in cycle length. *acd-5(lj122)* (**p=0.001) or the *acd-5(ok2657)* (****p<0.0001) double mutants showed significant increase in variability, similar to the *itr-1* single mutants. Error bars represent median and IQR. 75<N<130 (15–26 animals×5 cycles for each). See also *Figure 3—figure supplement 4*, showing comparison of cumulative distribution of cycle length and CV. (**F**) Effect of *acd-5* alleles on defecation cycle length of *pbo-4(ok538)* animals. DMP interval length (median and IQR). The Kruskal-Wallis test was significant with a post hoc Dunn's multiple comparison test indicating that only *acd-5(ok2657)* mutants had increased cycle length (****p<0.0001); *pbo-4* single (p=0.885) and double mutants (both p=1) were not significantly different from wild-type. *pbo-4(ok538)* significantly reduced the cycle length of *acd-5(ok2657)* animals (****p<0.0001). Error bars represent median and IQR. N=60 (12 animals × 5 cycles, wild-type), 40 (8×5 cycles, all other strains). (**G**) Schematic of the relationship between ACD-5, proton release and $Ca^{2+}$ oscillations. Release of protons at the luminal membrane via NHX channels (which could include PBO-4) and/or V-ATPase (of which VHA-6 is a component) results in decrease in luminal pH which inhibits the ACD-5 channel. Proton oscillations (curved line) from pH 4 to 6 activate the ACD-5 channel at pH 6 which in turn influences (but does not drive) *itr-1*-mediated $Ca^{2+}$ oscillations.

The online version of this article includes the following figure supplement(s) for figure 3:

**Figure supplement 1.** ACD-5 modifies intestinal $Ca^{2+}$ oscillations.

**Figure supplement 2.** Effect of *acd-5(ok2657)* overexpression on variation of defecation interval length and DMP.

**Figure supplement 3.** *acd-5* alleles alter the defecation phenotype of *itr-1* and *pbo-4* loss of function animals.

**Figure supplement 4.** Comparison of cumulative distribution of cycle length and coefficient of variation in *itr-1(sa73)* strains.

related. We investigated the subcellular localisation of ACD-3 and DEL-5 using translational fusions. In common with FLR-1::GFP (*Take-uchi et al., 1998*), mKate::DEL-5 and ACD-3::mKate2, where detectable, showed a diffuse distribution. Where expression levels were more variable, we were able to observe evidence of basolateral localisation, but we cannot exclude that DEL-5 and ACD-3 are also apically localised, since this could potentially be masked (*Figure 4A and B*). CRISPR-inserted fusions in the native genes were not detectable.

The presence of these three DEG/ENaC subunits on the basolateral membrane suggested that they might function as heteromers. We therefore asked whether co-expression of ACD-3, DEL-5 and FLR-1 in *Xenopus* oocytes would result in pH-sensitive currents. ACD-3, DEL-5, and FLR-1 expressed alone were insensitive to acidic pH (*Figure 4—figure supplement 2A,B*) and only the DEL-5 current was amiloride-sensitive, in fact showing slightly enhanced currents (*Figure 4—figure supplement 3A-C*), a phenomenon reported for some mammalian ASICs (*Adams et al., 1999*; *Li et al., 2011*). However, when we co-expressed FLR-1 with either DEL-5, ACD-3, or both, we observed a response to changes in pH from a holding pH 5 (pH$_{50}$ of 6.42), a profile resembling that previously reported for the TadNaCs, DEG/ENaC channels of the marine placozoa *Trichoplax adhaerens* (*Elkhatib et al., 2019*; *Figure 4C and D*, *Figure 4—figure supplement 4A,B*). We could not detect any currents in response to pH when expressing DEL-5 together with ACD-3 (*Figure 4—figure supplement 4C*). As FLR-1 has previously been proposed to localise at both the basolateral and luminal intestinal membrane (*Take-uchi et al., 1998*; *Kobayashi et al., 2011*), where it might form heteromers with ACD-5, we also co-expressed FLR-1 (ACD-3 or DEL-5) with ACD-5 in oocytes. The pH$_{50}$s did not change compared to the expression of ACD-5 alone (*Figure 4—figure supplement 5A,B*,). Co-expression can be difficult to interpret, due to the potential presence of a mix of homo-and heteromeric channels, which can confound whole-cell recordings (*Hesselager et al., 2004*). However, our results suggest that the other subunits do not influence ACD-5 channel function.

We also tested amiloride sensitivity of this heteromeric channel and found that it is insensitive to the blocker (*Figure 4E*), a phenomenon previously reported for a mutant DEGT-1 channel (*Fechner et al., 2021*). We next investigated ion selectivity, the heteromeric FLR-1/ACD-3/DEL-5 channel did not show a significant shift in $\Delta E_{rev}$ between the monovalent cations, however, it showed a negative shift in $\Delta E_{rev}$ when switching to a $CaCl_2$ solution with some current remaining (*Figure 4F–G*). This suggests that the channel is also marginally permeable for $Ca^{2+}$. Removal of $Na^+$ (by substituting with

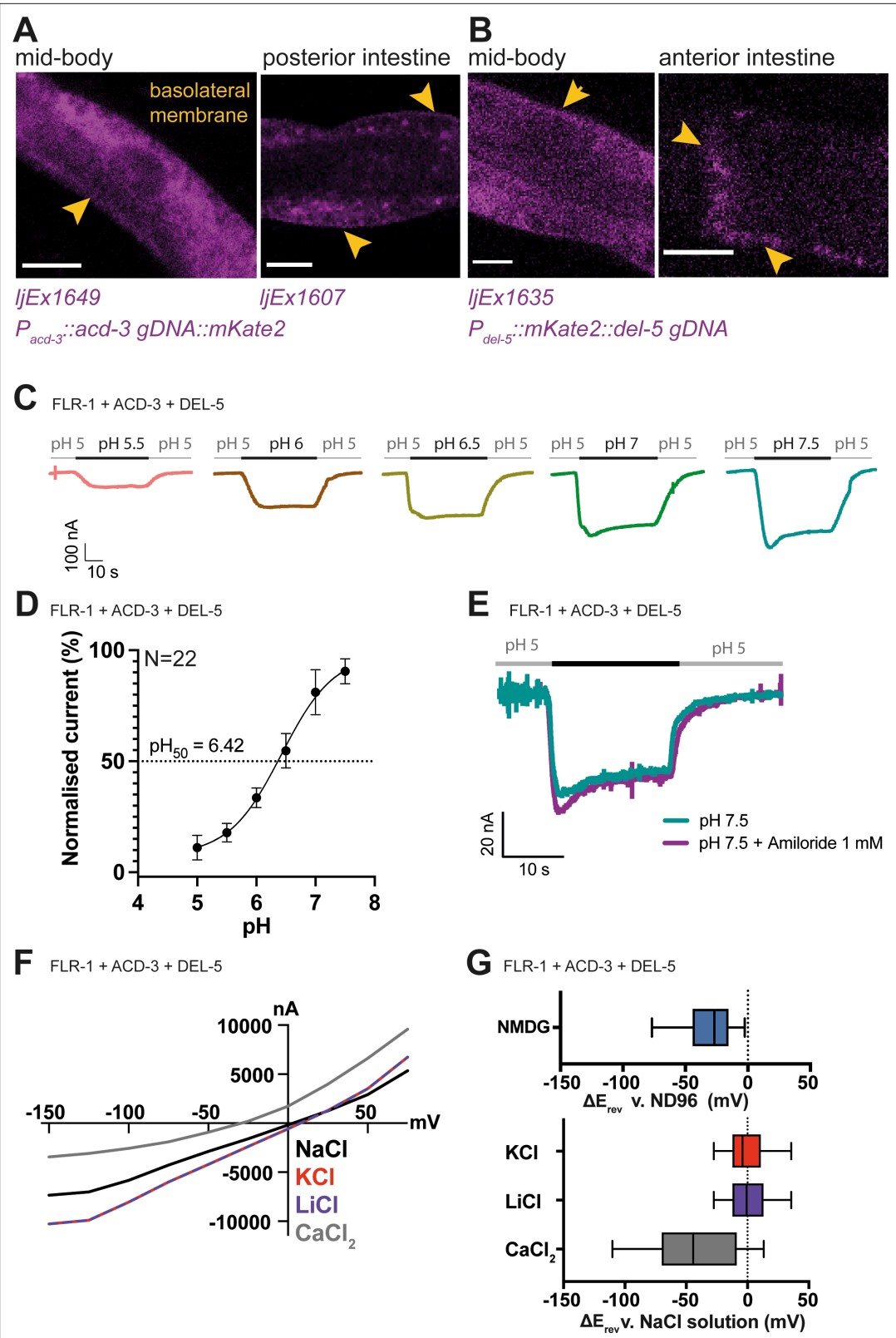

**Figure 4.** FLR-1 can form pH-sensitive heteromeric channels with ACD-3 and DEL-5. (**A**) ACD-3 is localised at the basolateral intestinal membrane. Localisation of mKate2-C-terminally tagged ACD-3 *ljEX1649* and *ljEx1607 (Pacd-3::acd-3 gDNA::mKate2)* (magenta) shows evidence of basolateral membrane localisation (yellow arrows). Shown are the mid-body (Left) and the posterior intestine (Right). Scale bar: 10μm. (**B**) DEL-5 is localised at the basolateral intestinal membrane. Localisation of mKate2-N-terminally tagged DEL-5 *ljEx1635 (Pdel-5:: mKate2::del-5 gDNA)* (magenta) shows evidence

*Figure 4 continued on next page*

*Figure 4 continued*

of basolateral and lateral membrane localisation (yellow arrows). Shown are the mid-body (Left) and the anterior intestine (Right). Scale bar: 10μm. See *Figure 5—figure supplement 3* for evidence that the fusions used in (**A**) and (**B**) rescue the defecation phenotypes of *acd-3/del-5* double mutants discussed later. (**C**) Representative traces from *Xenopus* oocytes injected with *flr-1*, *acd-3*, and *del-5* cRNA (250ng/μl of each construct) for the pH ranges pH 5.5–7.5 (gray bar represents perfusion time) from a holding pH of 5. Currents were recorded at a holding potential of –60mV and traces are baseline-subtracted and drift-corrected using the Roboocyte$^{2+}$ (Multichannels) software. (**D**) pH response curve (I/I$_{max}$ as percentage) covering the ranges pH 5–7.5 from a holding pH of pH 5. N=22. Mean ± SEM. Currents were recorded at a holding potential of –60mV, normalised to maximal currents and best fitted with the Hill's equation (Nonlin fit Log(inhibitor) versus normalised response–variable slope) in GraphPad Prism. (**E**) The FLR-1/ACD-3/DEL-5 heteromeric channel is insensitive to the blocker amiloride. Representative traces from *Xenopus* oocytes injected with *flr-1*, *acd-3*, and *del-5* cRNA (250ng/μl of each construct) perfused with pH 7.4 (black line) in the absence (cyan) or in the presence (purple) of 1mM amiloride. Currents were recorded at a holding potential of –60mV. (**F**) Current-voltage (I-V) curve representative trace and (**G**) change in reversal potential (median and IQR, Tukey method), ΔE$_{rev}$ when shifting from a ND96 to NMDG solution (top) (N=14 oocytes) or from a NaCl solution to KCl, LiCl, or CaCl$_2$ solution (bottom) (N=16 oocytes).

The online version of this article includes the following figure supplement(s) for figure 4:

**Figure supplement 1.** *acd-3* and *del-5* are expressed in the intestine.

**Figure supplement 2.** Characterisation of ACD-3, DEL-5, and FLR-1 homomeric channel acid-sensing properties.

**Figure supplement 3.** Characterisation of ACD-3, DEL-5, and FLR-1 homomeric channel amiloride-sensitivity.

**Figure supplement 4.** FLR-1 can form a pH-sensitive channel with ACD-3 or DEL-5.

**Figure supplement 5.** Co-expression with FLR-1, ACD-3, or DEL-5 does not change the pH$_{50}$ of ACD-5.

NMDG) also induced a negative shift in ΔE$_{rev}$ (*Figure 4G*). Thus, FLR-1, together with ACD-3 and/or DEL-5, appears to form a functional amiloride-insensitive acid-inhibited channel with different properties to ACD-5. These findings support the hypothesis that there is a second intestinal DEG/ENaC, an FLR-1, ACD-3, and DEL-5-containing heteromeric channel localised to the basolateral membrane.

## FLR-1, ACD-3, and DEL-5 function together, upstream of Ca$^{2+}$, to control DMP timing

*flr-1(ut11)* mutants show high variability of DMP cycle length, with many extremely short cycles (less than 30 s) (*Katsura et al., 1994*; *Take-uchi et al., 1998*; *Kwan et al., 2008*; *Take-uchi et al., 1998*). We created *flr-1(syb3521),* using CRISPR/Cas9 (*Dokshin et al., 2018*), truncating the N-terminus and predicted TM1 with the remaining gene being out of frame so no protein is generated (*Figure 1—figure supplement 2C,D*). As previously described for *flr-1(ut11)* and other alleles (*Katsura et al., 1994*; *Take-uchi et al., 1998*; *Kobayashi et al., 2011*), *syb3521* presented with a slow-growing, caloric restriction-like phenotype and were thus difficult to cross. Therefore, we used RNA interference (RNAi) by feeding (*Kamath et al., 2001*) to knock down *flr-1* in order to assess genetic interactions with the other potential subunit genes. We used *acd-3(ok1335)*, a truncation of most of the extracellular domain and TM2, and *del-5(lj138),* generated using CRISPR/Cas9 to delete the N-terminal fragment, the remainder of the gene being out of frame, so no protein is generated (*Figure 1—figure supplement 2E-H*).

As expected, *flr-1*(RNAi) animals also exhibited the caloric restriction phenotype. Like *flr-1(ut11)*, both *flr-1*(RNAi) and *flr-1(syb3521)* animals showed disruption of DMP, with a dramatic increase in variability, a mix of extremely short and long cycles, missed EMCs and missed or weak pBoc contractions (*Figure 5A–D* and *Figure 5—figure supplement 1*). Interestingly, *acd-3/del-5* double mutants phenocopied the *flr-1* deficient animals, showing similar defects in cycle length, EMC and pBoc defects (*Figure 5A–D*). *flr-1(RNAi)* did not significantly alter these phenotypes, suggesting that the three genes function together and further supporting the hypothesis that these three subunits form a heteromer. Again, we observed a compensatory effect, where a long cycle was likely to be followed by a short one (*Figure 5—figure supplement 2*). Neither of the *acd-3* or *del-5* single mutants exhibited such phenotypes, suggesting that they act redundantly. The defects observed for *acd-3/del-5* double mutants were rescued by expression of the *acd-3::mKate2* or *mKate2::del-5* fusions (*Figure 5—figure supplement 3*, *Table 1*), confirming that their functionality is retained and thus that the localisation observed in *Figure 4A* is functionally relevant.

Both the *flr-1* deficient animals and the *acd-3/del-5* double mutants frequently exhibited short <17 s cycles, seen for *unc-43(sa200)* CaMK II mutants and described as 'echos', since they are often incomplete (missing EMC) (*Teramoto and Iwasaki, 2006*; *Liu and Thomas, 1994*; *Figure 5E*). Although we

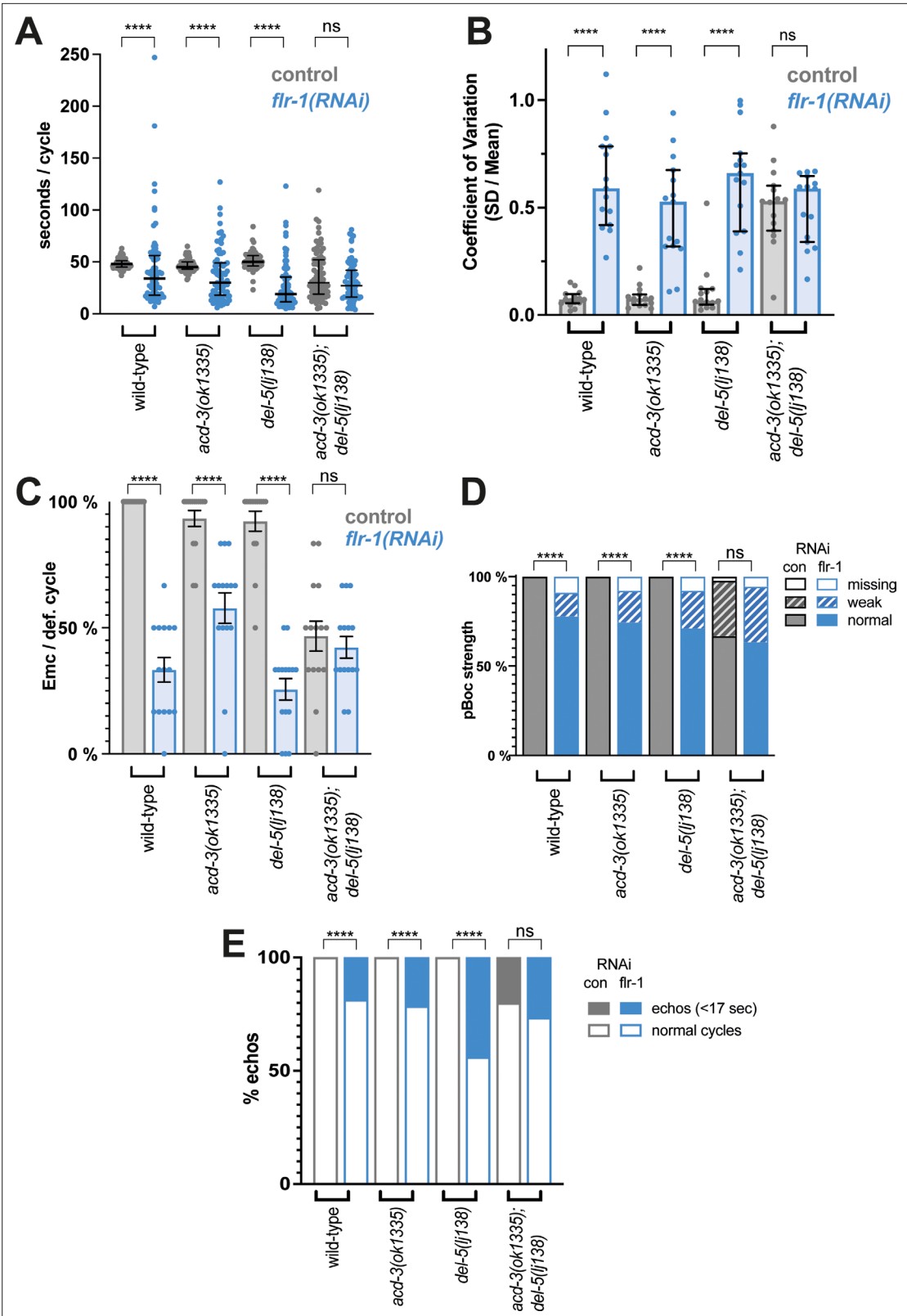

**Figure 5.** Knockdown of *flr-1* and mutations in *acd-3/del-5* results in arrhythmic DMP intervals, missed muscle contractions, and echos. FLR-1 or ACD-3/DEL-5 depletion alters cycle length and variability. (**A**) DMP interval and (**B**) coefficient of variation of animals on *flr-1*(RNAi) (blue) versus control RNAi (grey) assessed by a Mann-Whitney U-test (****p<0.0001, *acd-3(ok1335); del-5(lj138)* control vs. *flr-1 (RNAi)* p=0.0892for cycle length, p=0.5for CV). Error bars represent median and IQR. (**C**) Percentage of missed EMCs (mean and SEM) (****p<0.0001; *acd-3(ok1335); del-5(lj138)* control vs. *flr-1 (RNAi)*

*Figure 5 continued on next page*

*Figure 5 continued*

p=0.2535, Mann-Whitney). (**D**) Percentage of cycles displaying the pBoc characteristics indicated (****p<0.0001; *acd-3(ok1335); del-5(lj138)* control vs. *flr-1 (RNAi)* p=0.5861, Fisher exact test). (**E**) Percentage of 'echo' cycles, defined as cycles that last for under 17s (*Teramoto and Iwasaki, 2006*; *Liu and Thomas, 1994*) (****p<0.0001; *acd-3(ok1335); del-5(lj138)* control vs. *flr-1 (RNAi)* p=0.4516, Fisher exact test). N=13–15 animals for each genotype × 5 cycles scored for each animal.

The online version of this article includes the following figure supplement(s) for figure 5:

**Figure supplement 1.** *flr-1(ut11)* and *flr-1(syb3521)* mutants show similar phenotypes.

**Figure supplement 2.** Analysis of the defecation cycle of single and double mutants on *flr-1(RNAi)* reveals compensatory behaviour in the oscillator.

**Figure supplement 3.** Intestine-specific expression of *acd-3* or *del-5*, or expression of *acd-3* or *del-5* fluorophore fusions, rescues DMP defects of *acd-3/del-5* double mutants.

observed only intestinal expression for our transcriptional fusions (*Figure 4—figure supplement 1*), the http://www.cengen.org/ single-cell RNA sequencing database (*Hammarlund et al., 2018*, *Taylor et al., 2021*) indicates that *acd-3* and *del-5* are also expressed in DVB, a motor neuron involved in initiation of EMCs, so the observed phenotype could also potentially result from neuronal defects. As *Figure 5—figure supplement 3* shows, expression of either *acd-3* or *del-5* under an intestine-specific promoter significantly rescues all of the defects exhibited by *acd-3/del-5* double mutants, supporting the idea that the intestine is the site of function. This coincides with previous observations that *flr-1* is expressed only in the intestine, and that an intestinally expressed fusion rescues the *flr-1* mutant phenotype (*Take-uchi et al., 1998*). Previous research has shown that interrupting the $Ca^{2+}$ wave with heparin, an $IP_3$ receptor inhibitor, in parts of the intestine eliminates the aBoc and EMC following the pBoc, suggesting that the $Ca^{2+}$ wave is upstream of these steps (*Teramoto and Iwasaki, 2006*). The similarities with both the *itr-1* (highly variable, long cycles), and *unc-43* (echos) phenotypes suggests that the *acd-3/del-5* and *flr-1* mutations are likely to interfere with the intestinal $Ca^{2+}$ wave. As *Figure 6* shows, *flr-1(syb3521)* completely disrupted the rhythmicity of $Ca^{2+}$ oscillations that we normally observe in wild-type animals (see *Figure 6—figure supplement 1* and *Figure 6—figure supplement 2* for additional traces). Since we could not detect the weaker pBocs in our $Ca^{2+}$ recordings, we could not accurately correlate the properties of these $Ca^{2+}$ transients with the production of a pBoc contraction. Consistent with the huge variability in DMP interval and mix of normal, weak, and missed pBocs, we observed considerable variation and irregularity, making meaningful quantification difficult. Some recordings showed a constant low concentration; some showed higher magnitude, but irregular and noisy, $Ca^{2+}$ peaks; others showed regular, smaller peaks, with a shorter (<17 s) interval. Overall, both the basal and maximal $Ca^{2+}$ concentrations were significantly decreased (*Figure 6—figure supplement 3A,B*). Taken together, our data indicate that FLR-1 functions in a heteromer along with ACD-3 and/or DEL-5 and that this/these proton-gated channels, exposed to the pseudocoelom, play a major role in controlling the intestinal $Ca^{2+}$ transients that are so central to the DMP.

## FLR-1 is required for rhythmic intracellular intestinal and pseudocoelomic pH oscillations

Given the dramatic effect on $Ca^{2+}$ and the central role of $Ca^{2+}$ in driving cyclic events, we examined whether disrupting *flr-1* also disrupted pH oscillations. As *Figure 7* shows, *flr-1(RNAi)* impacted both intracellular and pseudocoelomic pH. Whereas basal intracellular pH was unaffected, the acidification that accompanies pBoc was not as extensive (*Figure 7A–C, G*). Likewise, consistent with the role of intracellular acidification in triggering proton release into the pseudocoelom (*Beg et al., 2008*; *Pfeiffer et al., 2008*), basal pseudocoelomic pH was unaffected, but the acidification (which triggers pBoc) was severely attenuated (*Figure 7D–G*). Interestingly, we also observed evidence of oscillations with a shorter periodicity, corresponding with the <17 s cycles and sorter interval $Ca^{2+}$ fluctuations. Our findings demonstrate that, in controlling intracellular $Ca^{2+}$, the FLR-1-containing channel also controls proton release and thus pseudocoelomic pH, explaining the defects in pBoc.

## Dysregulated intestinal proton signalling results in developmental delay and reduced fat storage

Many of the physiological changes associated with dysregulation of intestinal lumen pH reflect starvation, including developmental delay or arrest and reduced growth rate (*Benomar et al., 2020*;

**Table 1.** Intestine-specific expression of *acd-3* or *del-5*, or expression of *acd-3* or *del-5* fluorophore fusions, rescues DMP defects of *acd-3/del-5* double mutants.

Table showing p values for data shown in *Figure 5—figure supplement 3*.

| A. Interval length | p (Mann-Whitney) comparing with: | |
|---|---|---|
| Genotype | Wild-type | *acd-3(ok1335) del-5(lj138)* |
| *acd-3(ok1335) del-5(lj138)* | 0.0002 ns | – |
| +ljEx1634 [P$_{int}$::acd-3 cDNA] | 0.8441 ns | 0.0006 *** |
| +ljEx1637 [P$_{int}$::del-5 cDNA] | 0.3193 ns | 0.0003 *** |
| +ljEx1641 [P$_{int}$::acd-3 cDNA; P$_{int}$::del-5 cDNA] | 0.5833 ns | 0.0002 *** |
| +ljEx1640 [P$_{acd-3}$::acd-3 gDNA::mKate2] | 0.7638 ns | 0.1185 ns |
| +ljEx1635 [P$_{del-5}$::mKate2::del-5 gDNA] | <0.0001 **** | <0.0001 **** |

| B. CV of interval length | p (Mann-Whitney) comparing with: | |
|---|---|---|
| *acd-3(ok1335) del-5(lj138)* | <0.0001 **** | — |
| +ljEx1634 [P$_{int}$::acd-3 cDNA] | 0.0091 ** | <0.0001 **** |
| +ljEx1637 [P$_{int}$::del-5 cDNA] | 0.0141 * | <0.0001 **** |
| +ljEx1641 [P$_{int}$::acd-3 cDNA; P$_{int}$::del-5 cDNA] | 0.0620 ns | <0.0001 **** |
| +ljEx1640 [P$_{acd-3}$::acd-3 gDNA::mKate2] | 0.0162 * | 0.0008 *** |
| +ljEx1635 [P$_{del-5}$::mKate2::del-5 gDNA] | 0.0049 ** | <0.0001 **** |

| C. pBoC/def. cycle | p (Mann-Whitney) comparing with: | |
|---|---|---|
| *acd-3(ok1335) del-5(lj138)* | 0.0407 * | - |
| +ljEx1634 [P$_{int}$::acd-3 cDNA] | 1 ns | 0.0407 * |
| +ljEx1637 [P$_{int}$::del-5 cDNA] | 1 ns | 0.0407 * |
| +ljEx1641 [P$_{int}$::acd-3 cDNA; P$_{int}$::del-5 cDNA] | 1 ns | 0.0407 * |
| +ljEx1640 [P$_{acd-3}$::acd-3 gDNA::mKate2] | 1 ns | 0.0407 * |
| +ljEx1635 [P$_{del-5}$::mKate2::del-5 gDNA] | 1 ns | 0.0047 * |

| D. EMC/def. cycle | p (Mann-Whitney) comparing with: | |
|---|---|---|
| *acd-3(ok1335) del-5(lj138)* | <0.0001 **** | – |
| +ljEx1634 [P$_{int}$::acd-3 cDNA] | 0.2222 ns | <0.0001 **** |
| +ljEx1637 [P$_{int}$::del-5 cDNA] | 0.0407 * | 0.0005 *** |
| +ljEx1641 [P$_{int}$::acd-3 cDNA; P$_{int}$::del-5 cDNA] | 0.2222 ns | <0.0001 **** |
| +ljEx1640 [P$_{acd-3}$::acd-3 gDNA::mKate2] | 0.0058 ** | 0.0066 ** |
| +ljEx1635 [P$_{del-5}$::mKate2::del-5 gDNA] | 0.0978 ns | <0.0001 **** |

| E. Echos | p (Mann-Whitney) comparing with: | |
|---|---|---|
| *acd-3(ok1335) del-5(lj138)* | 0.0002 *** | – |
| +ljEx1634 [P$_{int}$::acd-3 cDNA] | 1 ns | 0.0012 ** |
| +ljEx1637 [P$_{int}$::del-5 cDNA] | 1 ns | 0.0012 ** |
| +ljEx1641 [P$_{int}$::acd-3 cDNA; P$_{int}$::del-5 cDNA] | 1 ns | 0.0012 ** |
| +ljEx1640 [P$_{acd-3}$::acd-3 gDNA::mKate2] | 0.0978 ns | 0.1025 ns |
| +ljEx1635 [P$_{del-5}$::mKate2::del-5 gDNA] | 1 ns | 0.0012 ** |

| F. pBoc strength (% normal) | p (Fisher exact) comparing with: | |
|---|---|---|
| *acd-3(ok1335) del-5(lj138)* | <0.0001 **** | – |

*Table 1 continued on next page*

*Table 1 continued*

| F. pBoc strength (% normal) | p (Fisher exact) comparing with: | |
| --- | --- | --- |
| +ljEx1634 [P<sub>int</sub>::acd-3 cDNA] | 1 ns | <0.0001 **** |
| +ljEx1637 [P<sub>int</sub>::del-5 cDNA] | 0.1205 ns | <0.0001 **** |
| +ljEx1641 [P<sub>int</sub>::acd-3 cDNA; P<sub>int</sub>::del-5 cDNA] | 0.4970 ns | <0.0001 **** |
| +ljEx1640 [P<sub>acd-3</sub>::acd-3 gDNA::mKate2] | 0.0007 *** | <0.0001 **** |
| +ljEx1635 [P<sub>del-5</sub>::mKate2::del-5 gDNA] | 1 ns | <0.0001 **** |

*Allman et al., 2009*; *Pfeiffer et al., 2008*). These characteristics resemble the phenotypes that have previously been described for *flr-1* mutants (*Take-uchi et al., 1998*). We noticed that the *acd-3/del-5* mutants also had an appearance suggesting caloric restriction, whereas the single *acd-3* and *del-5* mutants had a superficially normal appearance (*Figure 8—figure supplement 1*). However, we also wondered whether the disruption of pH homeostasis had consequences for development and metabolism. Therefore, we investigated the physiological consequences of these mutations in more detail. We quantified the time to develop, from laying of the egg to onset of egg-laying by the offspring (*Rollins et al., 2017*). Wild-type and the *acd-5(lj122)* mutants showed similar developmental timescales (median 74 and 74 hr), while there was a small but statistically significant delay in developmental timing for the *acd-5(ok2657)*, *acd-3(ok1335)*, and *del-5(lj138)* (median 75, 79, and 78 hr, respectively) (*Figure 8A*). By contrast, the *flr-1(syb3521)* single mutants and the *acd-3/del-5* double mutants showed a severe developmental delay, taking three times as long to reach adulthood (*Figure 8B*). A small fraction of these animals arrested and died as larvae or before egg-laying. When

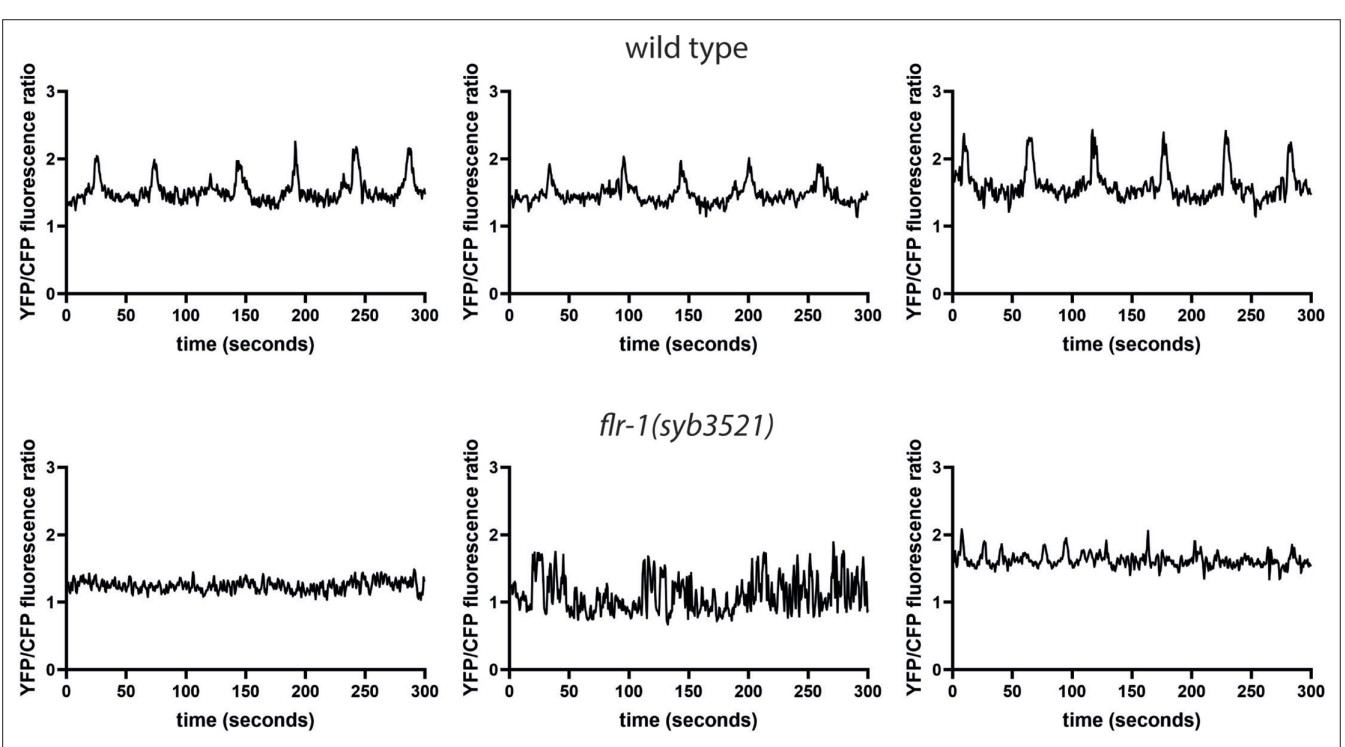

**Figure 6.** FLR-1 is required for rhythmic intestinal Ca²⁺ transients. Representative example traces of Ca²⁺ transients in the posterior intestinal cells (int9), recorded in freely moving animals expressing the ratiometric indicator D3cpv.

The online version of this article includes the following figure supplement(s) for figure 6:

**Figure supplement 1.** Additional intestinal Ca²⁺ traces for *flr-1(syb3521)*, demonstrating the disruption of rhythmicity.

**Figure supplement 2.** Additional intestinal Ca²⁺ traces for wild-type animals, demonstrating the rhythmicity of oscillations.

**Figure supplement 3.** FLR-1 is required for rhythmic intestinal Ca²⁺ oscillations.

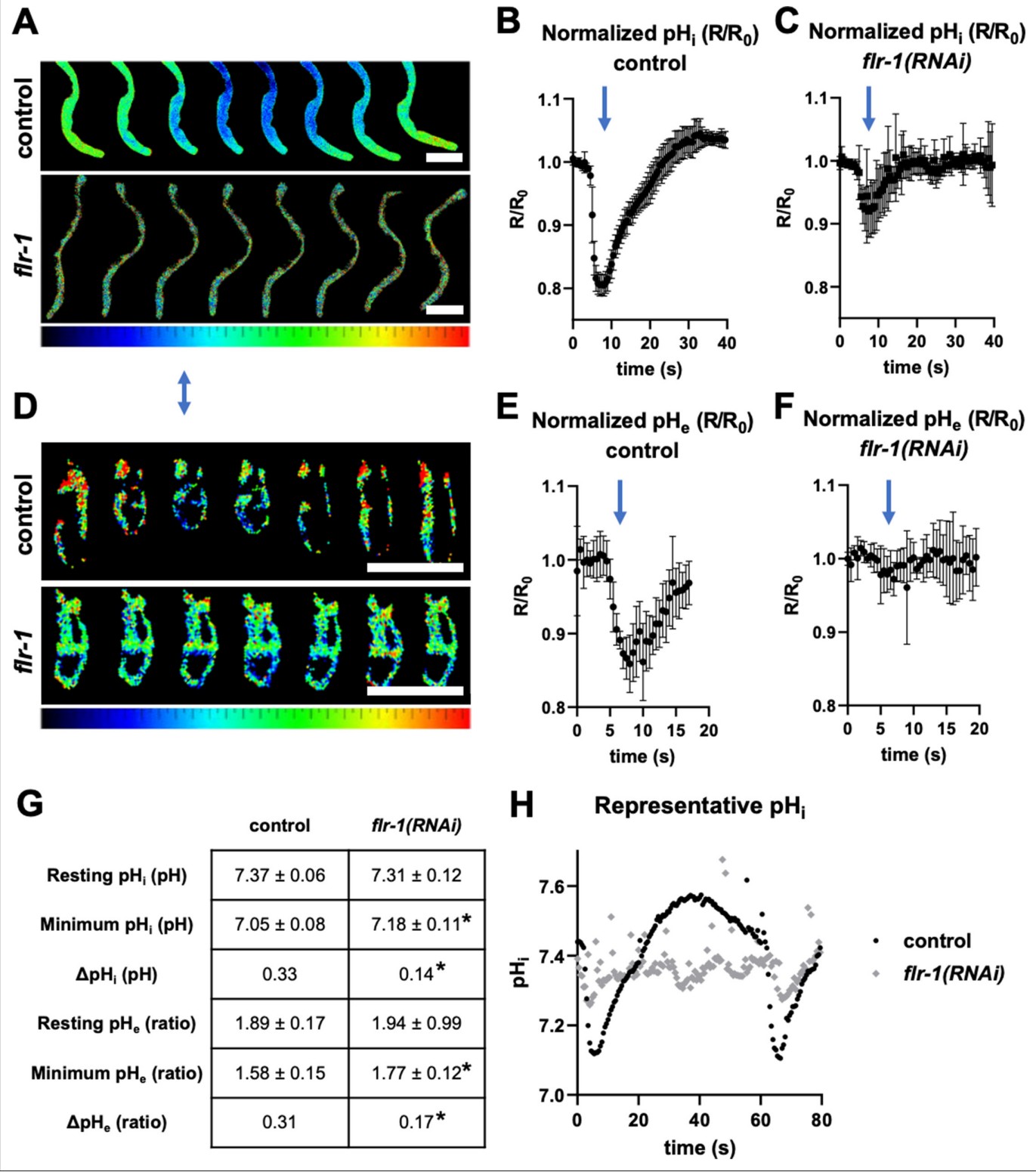

**Figure 7.** FLR-1 is required for rhythmic intracellular intestinal and pseudocoelomic pH oscillations. Dynamic ratiometric fluorescent imaging to measure pH in freely moving, defecating animals. (**A–C**) Animals expressing an intestinal cytoplasmic pH-sensitive GFP construct pHluorin, to measure $pH_i$. (**D–F**) Animals expressing an extracellular pH biosensor localised to the basolateral membrane of int9, to measure $pH_e$. (**A, D**) Composites of adjacent frames extracted from a representative experiment, with relative pH values mapped to a ratio palette, as shown. Images were acquired at 2Hz, and each series is time normalised to pBoc. Animals are oriented anterior upward. Blue arrow denotes pBoc. Scale bars indicate 100µm. (**B–C**) Plots of $R/R_0$ for pHi, time-normalised to pBoc, in control and *flr-1(RNAi)* treated worms, as indicated (N=6). (**E–F**) Plots of $R/R_0$ for pHe, time-normaliszed to pBoc, in control

*Figure 7 continued on next page*

*Figure 7 continued*

and *flr-1(RNAi)* treated worms, as indicated (N=6). Note that the x-axis differs for pHi and pHe traces. (**G**) Mean $pH_i$ ± SD (calculated using a Boltzmann calibration) or $pH_e$ (ratio) ± SD (raw YFP/CFP ratio, as calibration was not possible for this biosensor) for the resting and minimum observed values, as well as ΔpH (N=6). Control and respective *flr-1(RNAi)* means were statistically compared using two-tailed, unpaired t-tests (*p<0.05). (**H**) Single, extended $pH_i$ traces (80s) to highlight the arrhythmia, weak internal acidification, and 'echo' cycles observed in many *flr-1(RNAi)* animals.

we quantified the size of day-1 adults, we found that the *acd-3*, *acd-5*, and *del-5* single mutants show a wild-type length; by contrast, *flr-1(syb3521)* single mutants and the *acd-3/del-5* double mutants are significantly shorter (***Figure 8C***). These phenotypes are consistent with a role for FLR-1, ACD-3, and DEL-5 in development and growth.

Peptide absorption is dependent on proton gradients where nutrients are taken up against the gradient and greatly enhanced at a low extracellular pH (***Mackenzie et al., 1996***; ***Fei et al., 1994***; ***Liang et al., 1995***). In *C. elegans,* the proton gradient is known to drive nutrient uptake via the OPT-2/PEP-2, a proton-coupled oligopeptide transporter, which is also involved in fat accumulation and *opt-2* mutants show a decrease in intestinal fat deposits (***Nehrke, 2003***). Previous research revealed a link between DMP defects and aberrant fat metabolism, as well as between dysregulation of intestinal pH and reduced fat storage resulting in nutrient uptake deficiency (***Sheng et al., 2015***; ***Allman et al., 2009***). We therefore used Oil-Red O, a fat-soluble dye for staining triglycerides and lipoproteins, to observe fat distribution across tissues in the DEG/ENaC mutants (***Imanikia et al., 2019b***). Oil-Red O staining facilitates qualitative assessment of lipid distribution among tissues, but is not well suited to quantitative measurements (***Escorcia et al., 2018***; ***O'Rourke et al., 2009***). We found a clear reduction in intestinal fat accumulation, indicated by reduced Oil-Red O staining in *flr-1* single and *acd-3/del-5* double mutants, demonstrating aberrant fat metabolism (***Figure 8D***). This is consistent with the reduced growth rate and decreased body size and underlines the importance of proper regulation of the DMP, by the FLR-1/ACD-3/DEL-5 channel.

## Discussion

Using a combination of electrophysiology, behavioural analysis, genetic manipulations, and pH and $Ca^{2+}$ indicator imaging, we have obtained evidence for two acid-sensitive DEG/ENaC channels with distinct functions (***Figure 9***). One includes ACD-5, which localises to the luminal membrane and is the main proton sensing subunit of the channel. This proton sensor is required for maintenance and/or establishment of acidic luminal pH and yet only subtly influences the DMP, and the $Ca^{2+}$ oscillator. The second, composed of FLR-1, ACD-3, and/or DEL-5, localises to the basolateral membrane and is essential for timing and execution of DMP muscle contractions, with consequences for fat metabolism and growth.

ACD-5 and FLR-1-containing channels exhibit very different pH dependence and kinetics; ACD-5 is fully open at pH 6 and closed at pH 4, reflecting the range of pH oscillations in the lumen (***Allman et al., 2009***), whereas FLR-1-containing heteromers form a proton-inhibited cation leak channel, similar to ACD-1, TadNaC6 and vertebrate ASIC5 (***Wang et al., 2008***; ***Elkhatib et al., 2019***; ***Wiemuth and Gründer, 2010***). This presumably reflects their different physiological contexts, just as the pH dependence of human ENaCs reflects the pH range in epithelia where they are expressed (***Collier and Snyder, 2009***). Neither ACD-5 nor FLR-1 currents desensitise; the channel remains open for the duration of the pH stimulus (although we see some evidence of partial desensitisation of FLR-1 at certain pHs). This contrasts with the fast desensitisation of mammalian ASICs and is more analogous to the slower kinetics of ENaCs. It coincides well with our evidence that ACD-5 is actually required for establishment or maintenance of acidic luminal pH; that is, its activation ensures a return to the condition (pH~4) at which it is closed. By contrast, the FLR-1/ACD-3/DEL-5 heteromeric channel on the basolateral membrane is predicted to be constitutively open as the extracellular resting pH is around 7.35 (***Allman et al., 2009***).

All four intestinal DEG/ENaC subunits influence the DMP, but the degree of influence differs dramatically, providing us with mechanistic information about the relationships between extracellular and intracellular events. Research to date indicates that intestinal $IP_3$-dependent $Ca^{2+}$ signalling act as the master oscillator and is responsible for cycle length and timing. Disrupting $IP_3$ signalling results in a dramatic lengthening and disruption of rhythmicity of the DMP, and a direct relationship is seen

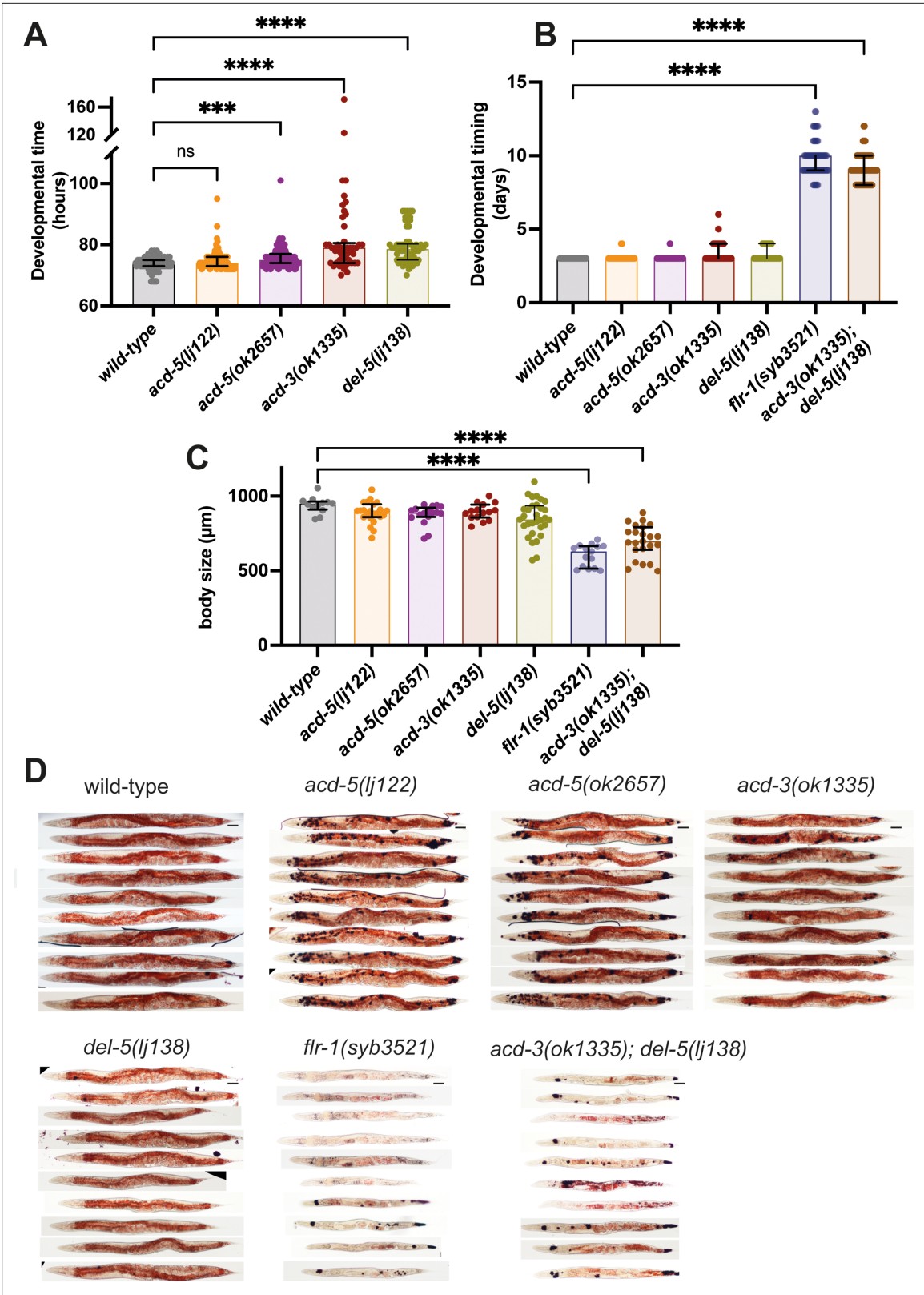

**Figure 8.** Consequences of disruptions of proton signalling in the intestine. (**A**) *acd-5(ok2657)*, *acd-3*, and *del-5* single mutants show a subtle developmental delay. Developmental timing (generation time) of DEG/ENaC single mutants in hours from egg laid to first egg laid as an adult. N=115, 102, 96, 49, and 62, in order on graph. Kruskal-Wallis test with Dunn's multiple comparison post hoc test, \*\*\*p=0.0004, \*\*\*\*p<0.0001. (**B**) *flr-1* single mutants and *acd-3/del-5* double mutants show a severe developmental delay. Developmental timing of DEG/ENaC single, double and triple mutants in

*Figure 8 continued on next page*

*Figure 8 continued*

days from egg-laid to day of onset of egg-laying. N=as in (**A**) for single mutants; 76, 92 for the remainder. (**C**) *flr-1* single mutants and *acd-3/del-5* double mutants are short. Quantification of body size of DEG/ENaC single, and double mutants. Each circle is an individual worm. N=12, 23, 17, 15, 31, and 15. Kruskal-Wallis test with Dunn's multiple comparison post hoc test, ****p<0.0001. Error bars indicate median and IQR. Each individual point is one animal. (**D**) *flr-1* single mutants and *acd-3/del-5* double mutants show aberrant fat metabolism. Fat storage and distribution of DEG/ENaC single and double mutants assessed by Oil-Red-O staining of 10 representative animals. Scale bar: 50µm.

The online version of this article includes the following figure supplement(s) for figure 8:

**Figure supplement 1.** *flr-1* and *acd-3/del-5* deficient animals exhibit a caloric restriction phenotype.

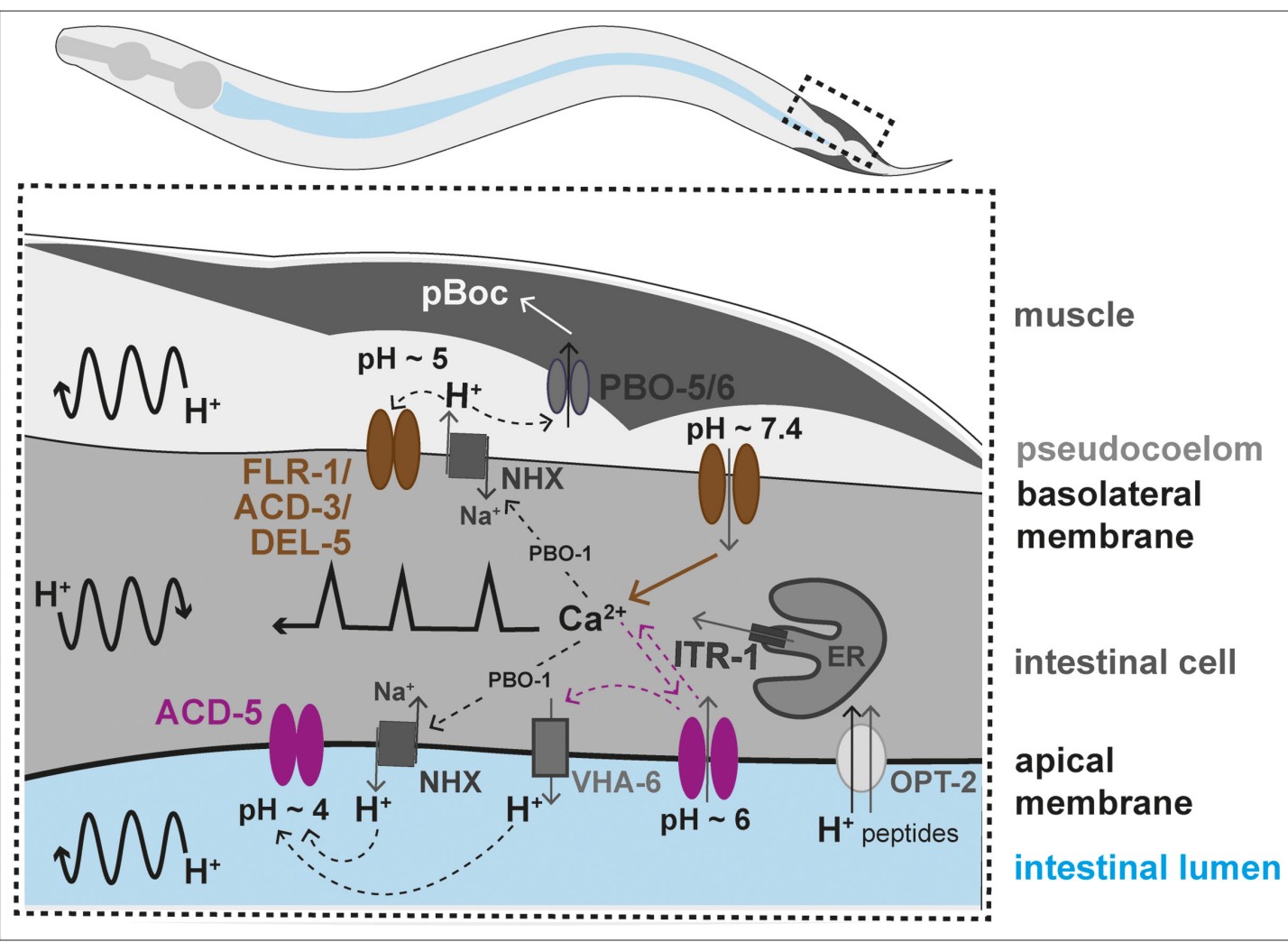

**Figure 9.** Working model of the two acid-sensing DEG/ENaCs in the *Caenorhabditis elegans* intestinal cells. $Ca^{2+}$ release from the endoplasmic reticulum (ER), via the $IP_3$ receptor, ITR-1, is responsible for the $Ca^{2+}$ oscillations transitioning from the posterior to the anterior part of the intestine (represented by the black line with sharp peaks). $Ca^{2+}$ also influences the activity $Na^+/H^+$ exchangers (NHX) via the CHP1 homolog PBO-1. $H^+$ oscillations occur in the intestinal cell, transitioning posteriorly, and pseudocoelom and intestinal lumen, transitioning anteriorly (represented by the curved black arrow lines). The FLR-1/ACD-3/DEL-5 containing channel localises to the basolateral intestinal membrane, it is likely open for much of the cycle, directly affecting $Ca^{2+}$ oscillations and ion homeostasis. The $Ca^{2+}$ transient triggers efflux of $H^+$ via NHXs (including PBO-4), decreasing pseudocoelomic pH and closing FLR-1/ACD-3/DEL-5, and $Ca^{2+}$ concentration returns to baseline. ACD-5 is localised at the apical membrane. Proton pumps such as NHXs and VHA-6 maintain the acidity in the lumen during most of the DMP (pH~4, ACD-5 is closed). Once a cycle, $H^+$ influx, through the $H^+$/dipeptide symporter OPT-2 and other unknown channels raises the luminal pH to about pH~6 (ACD-5 opens), the resulting cation-influx reactivates $H^+$ efflux (via VHA-6?). ACD-5 has limited influence on the $Ca^{2+}$ oscillations, masked by the dominant role of $IP_3$ signalling.

between the timing of Ca²⁺ fluctuations and the DMP (*Kwan et al., 2008*; *Walker et al., 2002*; *Espelt et al., 2005*; *Teramoto and Iwasaki, 2006*; *Dal Santo et al., 1999*). Our *acd-5* mutant data supports this hypothesis; the underlying Ca²⁺ fluctuations, and DMP rhythmicity, were only subtly affected, despite dramatic disruption of luminal pH, underlining the dominance of the Ca²⁺ oscillator. Where detectable, the MAT interval length was also not affected, suggesting that ACD-5 is not responsible for the timing of the proton wave, but acts downstream. This is supported by the phenotype of animals deficient in the V-ATPase, VHA-6; lumen pH remained more neutral, but robust Ca²⁺ oscillations were maintained, with only a small increase in cycle length (similar to *acd-5(ok2657)*) (*Allman et al., 2009*). This suggests a functional link between the two; a pH~6-activated cation channel and a presumed H⁺ pump whose effect would be to acidify the lumen, although the *vha-6* phenotype is more severe than *acd-5*, interfering with nutrient uptake and growth rate, indicating that there is not a complete overlap in their roles. Indeed, such a coupling has been proposed in Na⁺ uptake in the Zebrafish larva (*Zimmer et al., 2018*). We could speculate that the dominant mutation could result in interference with VHA-6 function, or alternatively it could be interfering with localisation or function of the other DEG/ENaC subunits (as we have shown that it can do for ACD-5 itself, when expressed in oocytes). Altering the subunit composition of the FLR-1-containing channels, rather than disrupting it completely, could result in such a subtle extension. Whatever the mechanism, it is clear that the Ca²⁺ oscillator is largely unperturbed. We do nevertheless see some influence; in particular, when the Ca²⁺ oscillator is disrupted, in *itr-1* deficient animals, the very long cycles observed are dependent on *acd-5*, revealing an otherwise constraining role. The Ca²⁺ oscillator is thus upstream, and/or largely independent, of lumen pH, and is the dominant force driving DMP rhythmicity.

In stark contrast to *acd-5* mutants, *flr-1* deficient animals exhibit a dramatic dysregulation of the DMP, with a combination of very long and very short cycles, underlining their importance for rhythmicity (as previous research showed; *Kobayashi et al., 2011*; *Kwan et al., 2008*; *Katsura et al., 1994*; *Take-uchi et al., 1998*), which is phenocopied by *acd-3/del-5* double mutants. The lack of a clear phenotype for *acd-3* and *del-5* single mutants suggests that they act redundantly, in a heteromer, along with FLR-1. This correlates well with our observation that FLR-1 pH-sensitive currents in *Xenopus* oocytes were dependent on the presence of one or other of these subunits, and that *acd-3/del-5* double mutant phenocopy other aspects of *flr-1* loss of function: EMC and pBoC defects, developmental timing, growth, and fat accumulation. The severity of disruption to the DMP is reminiscent of that seen when Ca²⁺ signalling is disrupted (*Kwan et al., 2008*; *Walker et al., 2002*; *Espelt et al., 2005*; *Dal Santo et al., 1999*). Indeed, we found that in *flr-1(syb3521)* animals, the rhythmicity and magnitude of intestinal Ca²⁺ transients, measured in the posterior cells, where the Ca²⁺ wave initiates, were completely disrupted. This indicates that the FLR-1/ACD-3/DEL-5 channel(s) is upstream of the Ca²⁺ oscillator, perhaps forming part of the 'master oscillator', linking extracellular proton concentration to the initiation of the intracellular Ca²⁺ wave that controls periodicity and execution of the DMP (see *Figure 9*).

Interestingly, in contrast to *itr-1*, the *flr-1* and *acd-3/del-5* deficient animals also exhibit very short cycles, and 'echos' of the DMP as well as missed or weak EMC or pBoc muscle contractions which has previously been described for the *unc-43* CaMK II mutant (*Teramoto and Iwasaki, 2006*) and *cmd-1* (calmodulin) RNAi (*Allman et al., 2013*), which again links the phenotypes of these mutants to the disturbance in intestinal Ca²⁺ signalling. These shorter cycles, for which we also saw evidence in both intracellular pH and Ca²⁺ oscillations, indicate a role for calmodulin, CaMK II and the FLR-1 channel in retarding a shorter timescale oscillator. Cycle variability, the other effect of *flr-1* or *acd-3/del-5* mutations, presumably represents a loss of synchronicity between semi-independent oscillators, each of which has an innate homeostatic rhythm—but is checked by a delay in reaching permissive conditions provided by another. A channel that is open for most of the cycle is consistent with a role in maintaining cellular excitability, and thereby underpinning intracellular Ca²⁺ signalling; closing briefly following proton release into the pseudocoelom would serve to terminate the signal. This model fits well with the relatively low currents observed for the FLR-1 channel (e.g., compared with ACD-5) and the temporal relationship observed (*Allman et al., 2009*), where Ca²⁺ concentration precisely follows the decline in pseudocoelom pH.

Notwithstanding the challenges associated with imaging freely moving animals, two important barriers to dissecting the relationships between events in the intestinal cells and their extracellular environments are the dependent crosstalk between signalling events and the fact that some components

are expressed on both the apical and basolateral membranes. PBO-1, for example, localises to both membranes, where it has distinct roles (*Allman et al., 2016*). Our data for ACD-3 and DEL-5 localisation to the basolateral membrane supports previous evidence for FLR-1 (*Kobayashi et al., 2011*). We did not detect apical localisation for ACD-3 or DEL-5 subunits. However, we did observe widespread, diffuse localisation within many cells, which might make any potential apical localisation difficult to distinguish, and *Take-uchi et al., 1998* reported apical localisation (*Take-uchi et al., 1998*), so we cannot exclude the possibility of a dual role at the two membranes. Basolateral localisation fits well with the pH dependence of the channel. However, an important remaining question is how exactly the FLR-1 channel influences $Ca^{2+}$ and calmodulin signalling, and the broad permeability of the channel does not help to shed light on this problem. Of key importance, and explaining why ACD-5 and FLR-1 have such distinct effects, may be the local nature of signalling events, with their activity thus specifically influencing components within their immediate vicinity.

We have shown that two acid-sensing DEG/ENaC channels with distinct properties play very different roles in the interplay between extracellular pH and intracellular $Ca^{2+}$. ACD-5 is required to maintain acidic lumen pH and has a subtle impact on the $Ca^{2+}$ oscillations that control DMP timing. In contrast, FLR-1, ACD-3, and/or DEL-5-containing channel is the essential link between pseudocoelomic pH and $Ca^{2+}$, and thus timing and execution of the DMP. DEG/ENaCs are conserved across the animal kingdom, and the ASICs and ENaC members are relevant to a broad range of medical conditions, and thus of significant therapeutic importance. Our data provide an insight into the relationship between the acid-sensing properties of these channels and their physiological role in epithelia. However, the family contains a diversity of members with distinct properties, and vastly expanded numbers in some species. Better understanding the relevance of channel properties to function will help us to better understand their role, and the role of acid-sensing, in other cellular contexts.

## Materials and methods

See Appendix 1 for Key resources table.

### *C. elegans* growth and maintenance

Standard techniques were used for *C. elegans* strain maintenance and genetic manipulations (*Brenner, 1974*). All experiments were performed on hermaphrodite worms grown on *Escherichia coli* OP50 at room temperature (22°C) and all animals were cultivated at 22°C unless otherwise stated. Strains with the temperature sensitive allele *itr-1(sa73)* were cultivated at the permissive temperature of 15°C to avoid sterility. Mutations were 6× outcrossed with the Bristol N2 wild-type, and transgenic strains were generated by microinjection of plasmid DNA (*Mello et al., 1991*). In order to generate the CRISPR mutations *acd-5(lj122)* or *del-5(lj138)*, the protocol established by the Mello lab was used (*Dokshin et al., 2018*). Guide RNAs (crRNA) and homology arms were ordered from IDT (Leuven Belgium) or Sigma-Aldrich (Merck Life Science UK Limited, Dorset, UK). For subcellular localisation of ACD-5 to the apical membrane localisation the strain KWN246 *pha-1(e2123) III; rnyEx133 [pKN114 (opt-2p::opt-2(aa1-412)::GFP)+pCL1 (pha-1+)]* was used. The following transgene was used for the free-moving *in vivo* $Ca^{2+}$ imaging experiments: *rnyEx109 [nhx-2p::D3cpv +pha-1(+)]* constructed by Keith Nehrke (*Wagner et al., 2011*; *Allman et al., 2009*). The following strains were provided by the *CAENORHABDITIS* GENETICS CENTER CGC, which is funded by NIH Office of Research Infrastructure Programs (P40 OD010440): RB2005 *acd-5(ok2657)I*; VC1047 *acd-3(ok1335) X*; JT73 *itr-1(sa73) IV*; RB793 *pbo-4(ok583) X*; KWN246 *pha-1(e2123) III; rnyEx133 [pKN114 (opt-2p::opt-2(aa1-412)::GFP)+pCL1 (pha-1+)]*; KWN190 *pha-1(e2123) III; him-5(e1490) V; rnyEx109 [nhx-2p::D3cpv +pha-1(+)]*. The following strain was created by SunyBiotech Co, Ltd (Fu Jian Province, China): PHX3521 *flr-1(syb3521)* (1416 bp deletion).

### Molecular biology

The transcriptional reporters *Pdel-5::GFP* and *Pacd-3::GFP* plasmids were a kind gift from Professor Kyuhyung Kim's lab (Daegu Gyeongbuk Institute of Science & Technology [DGIST], Korea) containing regulatory sequences 888 bp upstream of the *del-5* gene and 3085 bp upstream of the *acd-3* gene, respectively. For all other plasmids including the tagged proteins and sub-cloning of cDNA into KSM vector for *Xenopus* oocyte expression, the NEBuilder HiFi DNA Assembly Reaction Protocol was used

to assemble the vector and inserts using NEBuilder HiFi DNA Assembly Master Mix (Catalog #E2621L) and a vector:insert ratio 1:2 (0.5 pmol vector and the corresponding amount of insert using NEBio-calculator) see above. *C. elegans* cDNA was obtained from growing N2 wild-type animals on fifteen 6-cm NGM plates until the food was diminished, and subsequently extracted and purified using the TRIzol Direct-zol RNA Miniprep (Catalog #R2051, Zymo Research). cDNA was generated using the Invitrogen SuperScript III First-Strand Synthesis System (Catalog #18080051). Primers were designed using SnapGene 5.0.4. (Hifi-Cloning of two fragments) based on the cDNA gene sequence found on https://wormbase.org/, and ordered from Integrated DNA Technologies Inc (IDT) (Leuven Belgium) or Sigma-Aldrich (Merck Life Science UK Limited, Dorset, UK). The cDNA inserts were subcloned into the KSM vector under the control of the T7 promoter containing 3′ and 5′ untranslated regions (UTRs) of the *Xenopus* beta-globin gene and a poly(A) tail. The forward primer *agatctggttaccactaaac cagcc* and reverse primer *tgcaggaattcgatatcaagcttatcgatacc* were used to amplify the KSM vector. NEB $T_m$ calculator was used to determine annealing temperatures. For generation of mutations and deletions, the NEBaseChanger tool to generate primer sequences and an annealing temperature. The *acd-5(ok2657)* mutant cDNA was based on the *acd-5(ok2657)* mutant allele generated by *C. elegans* Gene Knockout Consortium (CGC).

## Confocal microscopy

Worms were mounted on 3% agar pads (in M9 (3 g $KH_2PO_4$, 6 g $Na_2HPO_4$, 5 g NaCl, and 1 M $MgSO_4$)) in a 3 μl drop of M9 containing 25 mM sodium azide ($NaN_3$, Sigma-Aldrich). Images were acquired using a Leica TCS SP8 STED 3× confocal microscope at 63× x, 40×, or 20× resolution and Z stacks and intensity profiles were generated using Fiji (ImageJ) (*Schneider et al., 2012*).

## Two-electrode voltage clamp in *Xenopus* oocytes

Linearised plasmid DNA was used as the template for *in vitro* RNA synthesis from the T7 promoter using the mMessage mMachine T3 Transcription Kit (Ambion #AM1348), producing 5′ Capped RNA. The reaction was incubating at 37°C for 2 hr, and the resulting RNA was purified by GeneJET RNA Cleanup and Concentration Micro Kit (Thermo Fisher Scientific #K0841) and eluted in 15 μl RNase-free water. *X. laevis* oocytes of at least 1 mm in size were obtained from EcoCyte Bioscience (Dortmund, Germany). They were de-folliculated by collagenase treatment and maintained in standard 1× ND96 solution (96 mM NaCl, 2 mM $MgCl_2$, 5 mM HEPES, 2 mM KCl, 1.8 mM $CaCl_2$, and pH 7.4). Oocytes were injected with 25 μl of cRNA solution at a total concentration of approximately 500 ng/μl using the Roboinject (MultiChannel Systems). Oocytes were kept at 16°C in 1× ND96 prior to TEVC. TEVC was performed 1–2 days post-injection at room temperature using the Roboocyte2 (MultiChannel Systems). *Xenopus* oocytes were clamped at −60 mV, using ready-to-use Measuring Heads from Multi-Channel Systems filled with 1.0 M KCl and 1.5 M potassium-acetate. All channels were tested using the Roboocyte2 (MultiChannel Systems). For all current-voltage steps (I-V) experiments, measurements were obtained in each solution once a steady-state current was achieved and the background leak current was subtracted. N represents different oocytes from independent experiments. All experiments were repeated at least three times on different days (i.e., using different batches of oocytes).

As millimolar concentrations of $Ca^{2+}$ and other divalent ions except $Mg^{2+}$ can block ASIC currents (*Paukert et al., 2004*), $Ca^{2+}$-free buffers were used for substitution experiments of monovalent cations adapted from a previous protocol (*Hardege et al., 2015*): 96 mM XCl, 1 mM $MgCl_2$, 5 mM HEPES, pH adjusted to 7.4 with XOH, where X was $Na^+$, $K^+$, or $Li^+$, respectively. For testing ion permeability for $Ca^{2+}$, a previous protocol was used (*Wang et al., 2008*) replacing $Na^+$ with equimolar $Ca^{2+}$. Previous research has shown that perfusing the oocytes with this $CaCl_2$ solution used here does not activate the oocyte endogenous $Ca^{2+}$-activated $Cl^-$ current (*Bianchi et al., 2004*; *Wang et al., 2008*). If necessary, D-Glucose was used to adjust osmolarity. The osmolarity was checked and confirmed to be within the error of 210 mosm (*Awayda and Subramanyam, 1998*). For testing pH sensitivity, 1× ND96 solutions were used; for solutions with a pH 5 or lower, MES was used instead of HEPES and adjusted with HCl. I-V relationships for ion selectivity were calculated by subtracting the background leak current in the presence of 500 μM amiloride for the ACD-5 homomeric channel, and the leak current at pH 5 for the FLR-1/ACD-3/DEL-5 heteromeric channel (which is closed at that pH) from the current observed in the absence of amiloride pH 7.4 in order to get the actual current. Actual current IV curves for each individual oocyte were fitted to a linear regression line and the x intercept was compared between

solutions to calculate an average reversal potential ($E_{rev}$). Reversal potential shift ($\Delta E_{rev}$) when shifting between pHs or from a NaCl to a KCl, LiCl, or $CaCl_2$ solution was calculated for each oocyte. In order to test the responses to pH, the channel-expressing *Xenopus* oocytes were perfused with 1× ND96 (using HEPES for buffering pH above 5.5 and MES for pH below 5), pH was adjusted with HCl and ranged from pH 7.4 (neutral pH of the ND96 solution) to pH 4. Background currents measured at pH 7.4 for ACD-5 and pH 5 for FLR-1-containing channels were subtracted from those measured during activation of the channels. For analysis, currents were normalised to maximal currents ($I/I_{max}$) and best fitted with the Hill equation (variable slope).

## Developmental timing of DEG/ENaC mutants and wild-type animals

The protocol described by *Rollins et al., 2017* was used. Briefly, animals were synchronised by allowing adults to lay eggs for 1 hr on an *E. coli* OP50 seeded 60 mm NGM plate. After hatching, animals were randomly picked onto individual 20 µl *E. coli* OP50 seeded NGM plates. Time (in hours and days) was counted from laid egg to egg-laying adulthood. The animals were kept at 20°C for the duration of the assay.

## Assessing and scoring defecation motor program

Defecation was assayed as previously described (*Thomas, 1990*). Briefly, animals were synchronised by allowing 10 day-1 hermaphrodites to lay for ~3 hr before picking them off and growing today-1 adult. On the day prior to the assay, transgenics were picked to a new plate (where applicable) and blinded. Following 2 min of acclimation on the microscope, five DMP cycles per animal were observed on a dissecting stereomicroscope at 50× magnification. The time elapsed between two pBocs was measured as one DMP cycle and success of the pBoc and EMC steps was recorded. 'echo' DMPs defined as cycles <17 s as previously described (*Liu and Thomas, 1994*). Strength of pBoc contractions was assessed visually. At least 15 animals were assayed across at least 3 days for five cycles each and wild-type and mutants were always scored alongside.

## RNA interference

RNAi feeding experiments were performed as described (*Kamath et al., 2003*; *Kamath and Ahringer, 2003*; *Kamath et al., 2001*). For behaviour assays, standard NGM plates were prepared, with 100 µg/ml carbenicillin and 1 mM IPTG added after autoclaving, and were poured 1–3 days before use (*Imanikia et al., 2019a*). The L4440 empty vector was used as a negative control and the X-6G10 (FLR-1) bacterial strain from the Ahringer *C. elegans* RNAi library (Source Bioscience) was used to knock down the *flr-1* gene. The bacterial strains were freshly grown overnight in LB +50 µg/ml ampicillin and 24 hr prior to each assay plates were seeded with 100 µl of overnight bacterial culture. Due to the severe developmental delay, animals were picked as L4s and scored as day-1 adults.

For pH imaging experiments, a slightly different protocol was followed: The control was HT115 lacking the *flr-1* targeting plasmid. Control and experimental bacterial cultures were grown overnight in LB broth with 50 µg/ml ampicillin and 15 µg/ml tetracycline, where appropriate, at 37°C, diluted 50-fold into LB without antibiotics and allowed to grow for 3 hr at 37°C, induced with 0.1 M isopropyl-thiogalactopyranoside (IPTG) at 37°C for 1 hr, concentrated fivefold, and seeded onto NGM agar plates containing 1 mM IPTG.

## Imaging and analysis of intestinal lumen pH

KR35 feeding, imaging, and image analysis were carried out as previously described (*Benomar et al., 2020*; *Bender et al., 2013*). Briefly, worms were raised to young adults on *E. coli* OP50. Prior to acquisition of videos, the animals were transferred to NGM plates supplemented with 10 µM KR35 and *E. coli* OP50 for 15–30 min, and then transferred and imaged on NGM plates without the fluorophore. All animals were treated with equivalent conditions on all days of imaging, including feeding the KR35 dye to animals of each genotype from the same source plate (one condition at a time) to avoid differences in dye concentration, immediately prior to imaging.

Imaging and image analysis were done as previously described (*Benomar et al., 2020*; *Bender et al., 2013*). Videos of free-moving animals fed with KR35 dyes were acquired on a Leica M165FC microscope using a Leica DFC3000G CCD Camera via the Leica Application Suite software (v4.40) (Leica Microsystems [Switzerland] Limited). Image sequences were acquired at 5× zoom, with 10×

gain, and at 10 frames per second. Illumination was via a Leica Kubler Codix source equipped with an Osram HXP 120 W lamp. Images were opened in Fiji (ImageJ), converted to 8-bit, scaled from 0 to 255 to a dynamic range of 10–120, and a rainbow RGB look-up table applied. Movies were converted to AVI using FIJI based on an approximately 30 s clip that corresponded to ~10–15 s before and after a MAT (where one occurred). Movies were acquired for ~2 min, to include 1–3 MATs. Movies were opened in ImageJ, and a circular ROI of 25×25 pixels was used to measure the fluorescence of the anterior-most intestine during a MAT. One to three measurements were taken per animal, and the data were transferred to Excel and GraphPad Prism for graphing and statistical analysis.

## Ca²⁺ imaging

$Ca^{2+}$ imaging was performed on freely moving animals. Transgenic day-1 adults expressing the genetically encoded calcium indicator Cameleon D3cpv in the intestine were imaged on standard 60 mm NGM plates for 5 min. Plates had been seeded with 5 µl *E. coli* OP50 the day before and grown at room temperature to yield a small lawn, spatially confining the worm's movement to ease tracking. Animals were picked onto these plates at least 10 min prior to recording and were allowed to acclimatise for at least 5 min following mounting the plate on the imaging microscope, visually confirming that they were defecating, and were manually tracked during video acquisition. Filter/dichroic pairs were as described previously (*Suzuki et al., 2003*). Images were recorded at 2 Hz, with 2×2 binning, using an iXon EM camera (Andor Technology), captured using IQ1.9 software (Andor Technology), and analysed using NeuronTracker, a Matlab (MathWorks) analysis script written by Ithai Rabinowitch (*Rabinowitch et al., 2013*). CFP/YFP fluorescence ratio was smoothed using a 2-frame running average to generate the individual traces shown. $Ca^{2+}$ "peaks" were identified using a 10-frame running average to identify the frame with maximum fluorescence and this information was used to align cycle transients for mean traces, and to calculate the time interval between peaks.

## Imaging of intracellular and pseudocoelomic pH

For dynamic pH imaging studies, transgenic animals expressing either pHluorin (*Miesenböck et al., 1998*) or EC-sensor (*Urra et al., 2008*) under the control of *C. elegans* intestinal promoters *nhx-2* (pan-intestinal) or *nhx-7* (int9) were used to visualise changes in intracellular and extracellular pH, respectively, as described (*Nehrke, 2003*; *Allman et al., 2013*). Gravid, fluorescent animals were placed on RNAi plates and allowed to lay eggs for approximately 8 hr, then removed. After 3 (pHluorin) or 4 (EC-sensor) days at 20°C, young adult progeny were transferred to 60 mm NGM-agarose plates seeded with OP50 bacteria for imaging. The plates were inverted onto the stage of a Nikon Eclipse TE2000-S microscope (Nikon Instruments, Melville, NY), and freely moving subjects were kept within the field of view manually via stage manipulation. Illumination was provided by a Polychrome IV monochromator (TILL Photonics, Germany). Emissions were collected on a high-speed charge-coupled device camera (PCO Imaging, Germany). TILLvisION software (TILL Photonics) was used for analysis. Data were compiled for presentation using Prism 8.0 (GraphPad Software, San Diego, CA).

Intracellular pH was assessed via dual excitation ratio spectroscopy, with sequential 10 ms excitation at 410 and 470 nm. Emissions were acquired at 535 nm, using 2×2 binning to increase signal to noise. Background subtraction was performed pixel-by-pixel at each excitation wavelength using images acquired from seeded plates lacking worms. Ratio images were further refined by thresholding, with the lowest intensity pixels (generally <2% of the data) set to zero. All images were processed using identical parameters. For quantitative analysis, fluorescence ratios were converted to pH using a Boltzmann equation and a calibration curve generated by *ex vivo* intestinal perfusion with a high $K^+$/ nigericin solution.

Extracellular pH was assessed via dual excitation ratio spectroscopy, with sequential 10 ms excitation at 435 and 490 nm. Emissions were acquired at 470 and 535 nm. Image processing was as described above, except ROIs were drawn around int9 to reduce background. ROI series started 10 frames before contraction and ended 10 frames after expulsion (30 frames minimum). The EC sensor measures (pH-sensitive YFP emissions/[pH-insensitive CFP emissions pH-sensitive CFP-to-YFP FRET]). The % FRET is not expected to fluctuate with pH. The EC sensor's fluorescent ratio in partly excised intestines of live worms was found to respond appropriately to pH changes in the superfusate. However, the fluorescent signal diminished quickly following excision, preventing proper calibration

of emission ratios to pH. Thus, extracellular pH values were represented as raw ratios, as described (*Allman et al., 2013*).

### Oil-Red-O staining and quantification of worm length

Oil-Red-O staining was performed as previously described (*Imanikia et al., 2019b*). Day-1 adults were rinsed with 1× phosphate-buffered saline (PBS), collected in 1.5 ml previously autoclaved Eppendorf tubes (to prevent worms from sticking to the tube), and washed twice with 1× PBS. After the final wash, supernatant was reduced to 120 µl and an equal volume of 2× MRWB (160 mM KCl, 40 mM NaCl, 14 mM $Na_2EGTA$, 1 mM spermidine-HCl, 0.4 mM spermine, 30 mM Sodium-PIPES pH 7.4, and 0.2% β-mercaptoethanol) was added along with 2% paraformaldehyde (PFA). Samples were incubated at room temperature on rocking stage for 1 hr. Worms were then allowed to settle by gravity and washed with 1× PBS to remove PFA. Supernatant was removed and to dehydrate the samples, 60% isopropanol was added and incubated at room temperature for 15 min. Oil-Red-O stock solution was prepared beforehand by dissolving the dye in isopropanol to a stock concentration of 0.5 mg/ml. Before use, Oil-Red-O was diluted with water to reach a final concentration of 60%, filtered, and 1 ml of dye added to each sample. Samples were then incubated on a rocking stage at room temperature overnight. Dye was then removed and worms washed with 1× PBS prior to imaging. Animals were mounted on 3% agarose pads and imaged at 10× or 20× magnification using an OLYMPUS BX41 with DIC optics connected to a NIKON Digital Sight. Image straightening was performed using ImageJ.

### Statistical analysis

Statistical analysis was performed in GraphPad Prism version 9.0.2 for macOS, GraphPad Software, San Diego, CA (https://www.graphpad.com/). Normality (if the data follows a normal/Gaussian distribution) was assessed by a Shapiro-Wilk test. If the data followed a normal distribution, a parametric test was deployed, if not, the nonparametric equivalent was chosen. Appropriate post hoc tests were always used for multiple comparisons (Bonferroni correction). The statistical tests used are indicated in each figure description.

## Acknowledgements

The authors are very grateful to Yee Lian Chew, Soudabeh Imanikia, Rebecca Taylor, and members of the Schafer, Taylor and de Bono (LMB), Beets and Timmerman (KU Leuven), and Pless (University of Copenhagen) labs, as well as Ewan St. John Smith (University of Cambridge) for helpful discussions and advice. The authors are grateful to the LMB Science and Support Facilities, in particular Ben Sutcliffe, Jonathan Howe, and Nick Barry from the Light Microscopy Team for their advice and help with microscopy, and Sue Hubbard, Mark Cussens, and Martyn Howard and the Team from Media and Glass for preparing solutions and NGM plates. The authors thank Kyuhyung Kim's lab (Daegu Gyeongbuk Institute of Science and Technology) for providing us with their *Caenorhabditis elegans* DEG/ENaC transcriptional reporter plasmids and sharing their unpublished expression data with us. Some strains were provided by the *Caenorhabditis* Genetics Center, which is funded by NIH Office of Research Infrastructure Programs (P40 OD010440), and some strains were generated by SunyBiotech (Fuzhou, China). Neurontracker was written by Ithai Rabinowitch. This work was funded by Medical Research Council MC-A023-5PB91 (William Schafer); National Institutes of Health R01NS110391 and R21DC015652 (William Schafer); Wellcome Trust WT103784MA (William Schafer); KU Center for Chemical Biology of Infectious Diseases P20GM113117 (Brian Ackley); National Institutes of Health R01AG067617 (Keith Nehrke). The funders had no role in study design, data collection and interpretation, or the decision to submit the work for publication.

## Additional information

### Funding

| Funder | Grant reference number | Author |
|---|---|---|
| Medical Research Council | MC-A023-5PB91 | William R Schafer |

| Funder | Grant reference number | Author |
|---|---|---|
| Wellcome Trust | WT103784MA | William R Schafer |
| KU Center for Chemical Biology of Infectious Diseases | P20GM113117 | Brian D Ackley |
| National Institutes of Health | R01AG067617 | Keith Nehrke |
| National Institutes of Health | R01NS110391 | William R Schafer |
| National Institutes of Health | R21DC015652 | William R Schafer |

The funders had no role in study design, data collection and interpretation, or the decision to submit the work for publication. For the purpose of Open Access, the authors have applied a CC BY public copyright license to any Author Accepted Manuscript version arising from this submission.

## Author contributions

Eva Kaulich, Conceptualization, Data curation, Formal analysis, Investigation, Methodology, Performed all experiments except calcium and pH imaging, Resources, Visualization, Writing - original draft, Writing – review and editing; Trae Carroll, Conceptualization, Data curation, Formal analysis, Investigation, Performed and analysed intracellular and pseudocoelomic pH experiments, Writing – review and editing; Brian D Ackley, Conceptualization, Data curation, Funding acquisition, Investigation, Methodology, Performed luminal pH experiments, Resources, Writing – review and editing; Yi-Quan Tang, Methodology, Supervision, Writing – review and editing, Provided significant technical guidance for electrophysiology; Iris Hardege, Methodology, Supervision, Writing – review and editing, Provided significant technical guidance for electrophysiology; Keith Nehrke, Conceptualization, Data curation, Formal analysis, Funding acquisition, Investigation, Methodology, Performed and analysed intracellular and luminal pH measurements, Resources, Supervision, Writing – review and editing; William R Schafer, Conceptualization, Funding acquisition, Methodology, Resources, Supervision, Writing – review and editing; Denise S Walker, Conceptualization, Data curation, Formal analysis, Investigation, Methodology, Performed calcium imaging experiments, Supervision, Writing - original draft, Writing – review and editing

## Author ORCIDs

Eva Kaulich http://orcid.org/0000-0002-0868-3702
Brian D Ackley http://orcid.org/0000-0002-1257-2407
William R Schafer http://orcid.org/0000-0002-6676-8034
Denise S Walker http://orcid.org/0000-0003-1534-1679

## Decision letter and Author response

Decision letter https://doi.org/10.7554/eLife.75837.sa1
Author response https://doi.org/10.7554/eLife.75837.sa2

# Additional files

## Supplementary files
• Transparent reporting form

## Data availability
All data generated or analysed during this study are included in the manuscript.

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

# Appendix 1

## Appendix 1—key resources table

| Reagent type (species) or resource | Designation | Source or reference | Identifiers | Additional information |
|---|---|---|---|---|
| Strain, strain background (*Escherichia coli*) | OP50 | *Caenorhabditis* Genetics Center (CGC) | OP50 | |
| Strain, strain background L4440 (empty vector) strain | RNAi control | Ahringer *C. elegans* RNAi library (Source Bioscience) | L4440 | |
| Strain, strain background, X-6G10 (FLR-1) strain | X-6G10 (FLR-1) | Ahringer *C. elegans* RNAi library (Source Bioscience) | *X-6G10* | |
| Peptide, recombinant protein | Alt-RO S.p. Cas9 Nuclease V3 | IDT | Cat#1081058 | |
| Chemical compound, drug | Oil Red O | Sigma Aldrich | Cat#O0625 | |
| Chemical compound, drug | Amiloride hydrochloride | Sigma Aldrich | Cat#A7410 | |
| Commercial assay, kit | NEBuilder HiFi DNA Assembly Master Mix | New England Biolabs Inc | Cat#E2621L | |
| Commercial assay, kit | TRIzol Direct-zol RNA Miniprep | Zymo Research | Cat#R2051 | |
| Commercial assay, kit | Invitrogen SuperScript III First-Strand Synthesis System | Invitrogen | Cat#18080051 | |
| Commercial assay, kit | Multisite Gateway Three-Fragment cloning system | Thermo Fisher Scientific | Cat#12537–023 | |
| Commercial assay, kit | mMessage mMachineTm T3 Transcription Kit | Ambion | Cat#AM1348 | |
| Commercial assay, kit | RNeasy Mini Kit | Qiagen | Cat#74,104 | |
| Biological sample (*Xenopus laevis* oocytes) | *X. laevis* oocytes | EcoCyte Bioscience (Dortmund, Germany) | | |
| Strain, strain background (*Caenorhabditis elegans*) | *acd-5(ok2657) I* | *Caenorhabditis* Genetics Center (CGC) | RB2005 | |
| Strain, strain background (*C. elegans*) | *acd-5(ok2657) I* | CGC | VC1047 | |
| Strain, strain background (*C. elegans*) | *itr-1(sa73) IV* | CGC | JT73 | |
| Strain, strain background (*C. elegans*) | *pbo-4(ok583) X* | CGC | RB793 | |
| Strain, strain background (*C. elegans*) | *flr-1(syb3521) X* | SunyBiotech Co., Ltd | PHX3521 | |
| Strain, strain background (*C. elegans*) | *flr-1(ut11) X* | **Take-uchi et al., 1998** | JC55 | |
| Strain, strain background (*C. elegans*) | *acd-5(lj122) I* | This study | AQ4667 | *C. elegans* strain generated using CRISP/Cas9 (for requesting strain please refer to the AQ number) |

*Appendix 1 Continued on next page*

*Appendix 1 Continued*

| Reagent type (species) or resource | Designation | Source or reference | Identifiers | Additional information |
|---|---|---|---|---|
| Strain, strain background (*C. elegans*) | *del-5(lj138) X* | This study | AQ4802 | *C. elegans* strain generated using CRISP/Cas9 (for requesting strain please refer to the AQ number) |
| Strain, strain background (*C. elegans*) | *del-5(lj138) X; acd-3(ok1335) X* | This study | AQ5188 | *C. elegans* strain (for requesting strain please refer to the AQ number) |
| Strain, strain background (*C. elegans*) | *acd-5(ok2657) I; itr-1(sa73) IV* | This study | AQ5048 | *C. elegans* strain (for requesting strain please refer to the AQ number) |
| Strain, strain background (*C. elegans*) | *acd-5(ok2657) I; pbo-4(ok583) X* | This study | AQ5050 | *C. elegans* strain (for requesting strain please refer to the AQ number) |
| Strain, strain background (*C. elegans*) | *acd-5(lj122) I; itr-1(sa73) IV* | This study | AQ5218 | *C. elegans* strain (for requesting strain please refer to the AQ number) |
| Strain, strain background (*C. elegans*) | *acd-5(lj122) I; pbo-4(ok583) X* | This study | AQ5074 | *C. elegans* strain (for requesting strain please refer to the AQ number) |
| Strain, strain background (*C. elegans*) | *acd-5(lj122) I; ljEx1248 [Pacd-5::acd-5 cDNA; Punc-122::GFP]* | This study | AQ4400 | *C. elegans* strain (for requesting strain please refer to the AQ number) |
| Strain, strain background (*C. elegans*) | *acd-5(ok2657); ljEx1249 [Pacd-5::acd-5 cDNA; Punc-122::GFP]* | This study | AQ4401 | *C. elegans* strain (for requesting strain please refer to the AQ number) |
| Strain, strain background (*C. elegans*) | *acd-5(ok2657) I; rnyEx109 [nhx-2p::D3cpv +pha-1(+)]* | This study | AQ5122 | *C. elegans* strain (for requesting strain please refer to the AQ number) |
| Strain, strain background (*C. elegans*) | *acd-5 (lj122) I; rnyEx109 [nhx-2p::D3cpv +pha-1(+)]* | This study | AQ5182 | *C. elegans* strain (for requesting strain please refer to the AQ number) |
| Strain, strain background (*C. elegans*) | *flr-1(syb3521) X; rnyEx109 [nhx-2p::D3cpv +pha-1(+)]* | This study | AQ5192 | *C. elegans* strain (for requesting strain please refer to the AQ number) |
| Strain, strain background (*C. elegans*) | *ljEx1470 [Pacd-5::acd-5(no stop) cDNA::mKate2; Punc-122::GFP]* | This study | AQ4872 | *C. elegans* strain (for requesting strain please refer to the AQ number) |
| Strain, strain background (*C. elegans*) | *ljEx1248[Pacd-5::acd-5 cDNA; Punc-122::GFP]* | This study | AQ4496 | *C. elegans* strain (for requesting strain please refer to the AQ number) |
| Strain, strain background (*C. elegans*) | *ljEx1249[Pacd-5::acd-5 cDNA; Punc-122::GFP]* | This study | AQ4497 | *C. elegans* strain (for requesting strain please refer to the AQ number) |
| Strain, strain background (*C. elegans*) | *ljEx1349 [Pdel-5::GFP; Punc-122::GFP]* | This study | AQ4652 | *C. elegans* strain (for requesting strain please refer to the AQ number) |
| Strain, strain background (*C. elegans*) | *ljEx1345 [Pacd-3::GFP; Punc-122::GFP]* | This study | AQ4648 | *C. elegans* strain (for requesting strain please refer to the AQ number) |
| Strain, strain background (*C. elegans*) | *ljEx1500 [Pacd-5::acd-5(ok2657); Punc-122::RFP]* | This study | AQ4934 | *C. elegans* strain (for requesting strain please refer to the AQ number) |
| Strain, strain background (*C. elegans*) | *ljEx1507 [Pges-1::acd-5(ok2657); Punc-122::RFP]* | This study | AQ4925 | *C. elegans* strain (for requesting strain please refer to the AQ number) |
| Strain, strain background (*C. elegans*) | *ljEx1607 [pEK337 (Pacd-3::acd-3 gDNA::mKate2); Punc-122::GFP]* | This study | AQ5194 | *C. elegans* strain (for requesting strain please refer to the AQ number) |
| Strain, strain background (*C. elegans*) | *ljEx1621 [pEK352 (Pdel-5::Nsig::mKate2::del-5 gDNA); Punc-122::GFP]* | This study | AQ5222 | *C. elegans* strain (for requesting strain please refer to the AQ number) |

*Appendix 1 Continued on next page*

*Appendix 1 Continued*

| Reagent type (species) or resource | Designation | Source or reference | Identifiers | Additional information |
|---|---|---|---|---|
| Strain, strain background (*C. elegans*) | pha-1(e2123) III; rnyEx133 [pKN114 (opt-2p::opt-2(aa1-412)::GFP)+pCL1 (pha-1+)]; | CGC *Wagner et al., 2011*; *Allman et al., 2009* | KWN246 | |
| Strain, strain background (*C. elegans*) | pha-1(e2123) III; him-5(e1490) V; rnyEx109 [nhx-2p::D3cpv +pha-1(+)] | CGC *Wagner et al., 2011*; *Allman et al., 2009* | KWN190 | |
| Strain, strain background (*C. elegans*) | pha-1(e2123ts)III, rnyEx006 [pIA5nhx-2 (Pnhx-2::pHluorin)], pCL1 (pha-1+) | *Allman et al., 2009* | KWN26 | |
| Strain, strain background (*C. elegans*) | pha-1(e2123ts)III, him-5(e1490)V, rnyEx116 [pKT82 (Pnhx-7::EC sensor), pCL1 (pha-1+)] | This study | KWN197 | *C. elegans* strain (for requesting strain please refer to the KWN number) |
| Sequence-based reagent | Pacd-5 F | This study | PCR primer for generating pEK165 | Pacd-5 F ggggacaactttgtatagaaaagttgtaagaaaaattatcacatttttgtagatgaaac |
| Sequence-based reagent | Pacd-5 R | This study | PCR primer for generating pEK165 | Pacd-5 R ggggactgctttttttgtacaaacttgtcctggaacaactccactttcaaacctg |
| Sequence-based reagent | acd-5 cDNA_KSM_F | This study | PCR primer for generating pEK171 | acd-5 cDNA_KSM_F ttgggcccctcgaggtcgacatgcgacgcgtaagaaacc |
| Sequence-based reagent | acd-5 cDNA_KSM_R | This study | PCR primer for generating pEK171 | acd-5 cDNA_KSM_R ctccattcgggtgttcttgattatgcttcatgtatcacagctggc |
| Sequence-based reagent | acd-5(ok2657) mut_F | This study | PCR primer for generating pEK172 | acd-5(ok2657) mut_F tgtcaagaacacccgaatgg |
| Sequence-based reagent | acd-5(ok2657) mut_R | This study | PCR primer for generating pEK172 | acd-5(ok2657) mut_R cattcatatgcggaatcgtc |
| Sequence-based reagent | Pacd-5_acd-5_no stop_F | This study | PCR primer for generating pEK190 | Pacd-5_acd-5_no stop_F ccaagctcggacaccgttaatccaattactcttcaacatccctacatgc |
| Sequence-based reagent | Pacd-5_acd-5_no stop_R | This study | PCR primer for generating pEK190 | Pacd-5_acd-5_no stop_R ctccttgatgagctcggacattttttctaccggtactttctgatctactccgccgactt |
| Sequence-based reagent | mKate2_Fragment_F | This study | PCR primer for generating pEK190 | mKate2_Fragment_F gtcggcggagtagatcagaaagtaccggtagaaaaaatgtccgagctcatcaaggagaac |
| Sequence-based reagent | mKate2_Fragment_R | This study | PCR primer for generating pEK190 | mKate2_Fragment_R gatgttgaagagtaattggattaacggtgtccgagcttgga |
| Sequence-based reagent | C27C12.5b_acd-3_KSM_F | This study | PCR primer for generating pEK207 | C27C12.5b_acd-3_KSM_F ccctcgaggtcgacggatgactgaaacttcaaattgctccag |
| Sequence-based reagent | C27C12.5b_acd-3_KSM_R | This study | PCR primer for generating pEK207 | C27C12.5b_acd-3_KSM_R cttagagactccattcgggtttagaaatcacaatttccgagatacacagaatttctttt |
| Sequence-based reagent | F02D10.5_flr-1_KSM_F | This study | PCR primer for generating pEK215 | F02D10.5_flr-1_KSM_F agcttgatatcgaattcctgatggaaacggagacggaaagtg |
| Sequence-based reagent | F02D10.5_flr-1_KSM_R | This study | PCR primer for generating pEK215 | F02D10.5_flr-1_KSM_R ttcttgaggctggtttagtgtcaaattaattgtgatttgaatatggaggatgttgaaact |
| Sequence-based reagent | F59F3.4_del-5_KSM_F | This study | PCR primer for generating pEK228 | F59F3.4_del-5_KSM_F cttgatatcgaattcctgcaatgacgagtgtctcgtttggt |
| Sequence-based reagent | F59F3.4_del-5_KSM_R | This study | PCR primer for generating pEK228 | F59F3.4_del-5_KSM_R gtttagtggtaaccagatctttaaaaatcattcataggcatattttggtgaatgct |
| Sequence-based reagent | acd-5_cDNA_attB1_F | This study | PCR primer for generating pEK262 | acd-5_cDNA_attB1_F ggggacaagtttgtacaaaaaagcaggcttaatgcgacgcgtaagaaacct |
| Sequence-based reagent | acd-5_cDNA_attB1_R | This study | PCR primer for generating pEK262 | acd-5_cDNA_attB1_R ggggaccactttgtacaagaaagctgggtttttatgcttcatgtatcacagctggc |
| Sequence-based reagent | acd-5(ok2657) mut_F | This study | PCR primer for generating pEK286 | acd-5(ok2657) mut_F tgtcaagaacacccgaatgg |
| Sequence-based reagent | acd-5(ok2657) mut_R | This study | PCR primer for generating pEK286 | acd-5(ok2657) mut_R cattcatatgcggaatcgtc |
| Sequence-based reagent | acd-5(ok2657) mut_F | This study | PCR primer for generating pEK287 | acd-5(ok2657) mut_F tgtcaagaacacccgaatgg |

*Appendix 1 Continued on next page*

*Appendix 1 Continued*

| Reagent type (species) or resource | Designation | Source or reference | Identifiers | Additional information |
|---|---|---|---|---|
| Sequence-based reagent | acd-5(ok2657) mut_R | This study | PCR primer for generating pEK287 | acd-5(ok2657) mut_R cattcatatgcggaatcgtc |
| Sequence-based reagent | flr-1_F | This study | PCR primer for generating pEK288 | flr-1_F ggtaccgagctcggatccacatggaaacggagacggaaagtga |
| Sequence-based reagent | flr-1_noStop_R | This study | PCR primer for generating pEK288 | flr-1_noStop_R tcgaattccaccacactggaaattaattgtgatttgaatatgga ggatgttgaaact |
| Sequence-based reagent | acd-3_F | This study | PCR primer for generating pEK289 | acd-3_F ggtaccgagctcggatccacatgactgaaacttcaaattgctccagc |
| Sequence-based reagent | acd-3_noStop_R | | PCR primer for generating pEK289 | acd-3_noStop_R tcgaattccaccacactggagaaatcacaatttccgagat acacagaatttct |
| Sequence-based reagent | del-5_F | This study | PCR primer for generating pEK290 | del-5_F ggtaccgagctcggatccacatgacgagtgtctcgtttggt |
| Sequence-based reagent | del-5_noStop_R | This study | PCR primer for generating pEK290 | del-5_noStop_R tcgaattccaccacactggaaaaatcattcataggcatattttt ggtgaatgct |
| Sequence-based reagent | acd-5_pcDNA_F | This study | PCR primer for generating pEK291 | acd-5_pcDNA_F ccactagtccagtgtggtggatgcgacgcgtaagaaacc |
| Sequence-based reagent | acd-5_pcDNA_R | This study | PCR primer for generating pEK291 | acd-5_pcDNA_R actgtgctggatatctgcagttatgcttcatgtatcacag ctggcg |
| Sequence-based reagent | ok2657_v1_F | This study | PCR primer for generating pEK292 | ok2657_v1_F aggtcaagacaattctgcag |
| Sequence-based reagent | ok2657_v1_R | This study | PCR primer for generating pEK292 | ok2657_v1_R ttcatatgcggaatcgtcc |
| Sequence-based reagent | KSM_F | This study | PCR primer for generating any KSM backbone containing plasmid | KSM_F agatctggttaccactaaaccagcc |
| Sequence-based reagent | KSM_R | This study | PCR primer for generating any KSM backbone containing plasmid | KSM_R tgcaggaattcgatatcaagcttatcgatacc |
| Sequence-based reagent | Pacd-3_F | This study | PCR primer for generating pEK337 | Pacd-3_F ggaaacagctatgaccatgtttatcgcaaaatttcgtacatcaaatat tcaatgattcag |
| Sequence-based reagent | acd-3_mKate2_R | This study | PCR primer for generating pEK337 | acd-3_mKate2_R tccttgatgagctcggacatcatgaaatcacaatttccga gatacacagaatttctt |
| Sequence-based reagent | mKate2_acd-3_F | This study | PCR primer for generating pEK337 | mKate2_acd-3_F tcggaaattgtgatttcatgatgtccgagctcatcaagga gaac |
| Sequence-based reagent | mKate2_R | This study | PCR primer for generating pEK337 | mKate2_R ccgtacatgaaggaggtggcg |
| Sequence-based reagent | mKate2_unc-54 3′URT _R | This study | PCR primer for generating pEK337 | mKate2_unc-54 3′URT _R accggcgctcagttggaattttaacggtgtcc gagcttgga |
| Sequence-based reagent | unc-54 3′URT_mKate2_F | This study | PCR primer for generating pEK337 | unc-54 3′URT_mKate2_F ccaagctcggacaccgttaaaattccaactga gcgccgg |
| Sequence-based reagent | Pdel-5_F | This study | PCR primer for generating pEK337 | Pdel-5_F aggaaacagctatgaccatgctaaactatcaaaatacacgaaagttat ctaaaaacctc |
| Sequence-based reagent | del-5_mKate2_R | This study | PCR primer for generating pEK337 | del-5_mKate2_R gttctccttgatgagctcggacatcataaaatcattcata ggcatattttggtgaatgc |
| Sequence-based reagent | mKate2_del-5_F | This study | PCR primer for generating pEK338 | mKate2_del-5_F tgcctatgaatgattttatgatgtccgagctcatcaagga gaac |
| Sequence-based reagent | mKate2_R | This study | PCR primer for generating pEK338 | mKate2_R ccgtacatgaaggaggtggcg |
| Sequence-based reagent | mKate2_unc-54 3′URT _R | This study | PCR primer for generating pEK338 | mKate2_unc-54 3′URT _R accggcgctcagttggaattttaacggtgtcc gagcttgga |
| Sequence-based reagent | unc-54 3′URT_mKate2_F | This study | PCR primer for generating pEK338 | unc-54 3′URT_mKate2_F ccaagctcggacaccgttaaaattccaactga gcgccgg |
| Sequence-based reagent | CD.Cas9.VZPJ2357. AN+AltR2 | This study | Guide RNA for CRISPR deletion *acd-5(lj122)I* | CD.Cas9.VZPJ2357.AN gtttgatctgaaagatatca+AltR2 |

*Appendix 1 Continued on next page*

*Appendix 1 Continued*

| Reagent type (species) or resource | Designation | Source or reference | Identifiers | Additional information |
|---|---|---|---|---|
| Sequence-based reagent | Ce.Cas9.ACD-5.1.AK+AltR2 | This study | Guide RNA sequence for CRISPR deletion *acd-5(lj122)I* | Ce.Cas9.ACD-5.1.AK tttgtataacgacggaccga+AltR2 |
| Sequence-based reagent | ssODN_*acd-5* | This study | Repair template for CRISPR deletion *acd-5(lj122)I* | ssODN_acd-5 aacctattattcacaccgttcagaaacctctca ctaggatcaaataacacgtggaagggctgcatcaatg |
| Sequence-based reagent | Ce.Cas9.DEL-5.1.AE+Alt2 | This study | Guide RNA sequence for CRISPR deletion *del-5(lj138)X* | Ce.Cas9.DEL-5.1.AE accaatcattattttcgagc+Alt2 |
| Sequence-based reagent | Ce.Cas9.DEL-5.1.AD+Alt2 | This study | Guide RNA sequence for CRISPR deletion *del-5(lj138)X* | Ce.Cas9.DEL-5.1.AD caaagtcttgatccagctcc+Alt2 |
| Sequence-based reagent | del-5_ssODN | This study | Repair template for CRISPR deletion *del-5(lj138)X* | del-5_ssODN gactctcaatttcaaagtcttgatccagctccttaatgtcatga tgggtgtgatgaatacaccaatttaa |
| Sequence-based reagent | *acd-5_Fwd* | This study | PCR primer for genotyping *acd-5(lj122)I or acd-5(ok2657)I* | *acd-5_Fwd* cgcagctagagtttcacagc |
| Sequence-based reagent | *acd-5_Rev* | This study | PCR primer for genotyping *acd-5(lj122)I or acd-5(ok2657)I* | *acd-5_Rev* cagagctttaacattgagatgcc |
| Sequence-based reagent | *del-5(lj138)_F* | This study | PCR primer for genotyping *del-5(lj138)X* | *del-5(lj138)_F* aatgacgagtgtctcgtttg |
| Sequence-based reagent | *del-5(lj138)_R* | This study | PCR primer for genotyping *del-5(lj138)X* | *del-5(lj138)_R* cgttgattctcataaaactggg |
| Sequence-based reagent | *acd-3(ok1335)_F* | This study | PCR primer for genotyping *acd-3(ok1335)X* | *acd-3(ok1335)_F* catgaaagtactaacttccagattcg |
| Sequence-based reagent | *acd-3(ok1335)_R* | This study | PCR primer for genotyping *acd-3(ok1335)X* | *acd-3(ok1335)_R* gagatccatggatacttccg |
| Sequence-based reagent | *flr-1(syb3521)_F* | This study | PCR primer for genotyping *flr-1(syb3521)X* | *flr-1(syb3521)_F* tcgagggcacaagctcataaa |
| Sequence-based reagent | *flr-1(syb3521)_R* | This study | PCR primer for genotyping *flr-1(syb3521)X* | *flr-1(syb3521)_R* cccaaccaaaaccatttccact |
| Sequence-based reagent | *itr-1(sa73)_F* | This study | PCR primer for genotyping *itr-1(sa73)IV* | *itr-1(sa73)_F* cgagctttcgattcgggaga |
| Sequence-based reagent | *itr-1 (sa73)_R* | This study | PCR primer for genotyping *itr-1(sa73)IV* | *itr-1 (sa73)_R* tgatcaacacacggcaggtaa |
| Sequence-based reagent | *pbo-4(ok583)_F* | This study | PCR primer for genotyping *pbo-4(ok583)X* | *pbo-4(ok583)_F* actagatgagagttggcgaga |
| Sequence-based reagent | *pbo-4(ok583)_R* | This study | PCR primer for genotyping *pbo-4(ok583)X* | *pbo-4(ok583)_R* agtcgtgtggtaaagctccg |
| Recombinant DNA reagent | Plasmid: *pDEST Pacd-5::mKate2* | This study | pEK165 | Plasmid generated for generating transgenic *C. elegans* strains (for requesting plasmid please refer to the pEK number) |
| Recombinant DNA reagent | Plasmid: *KSM acd-5 cDNA* | This study | pEK171 | Plasmid generated for generating cRNA for the expression of channels in *Xenopus oocytes* (for requesting plasmid please refer to the pEK number) |
| Recombinant DNA reagent | Plasmid *KSM acd-5(ok2657) cDNA* | This study | pEK172 | Plasmid generated for generating cRNA for the expression of channels in *Xenopus oocytes* (for requesting plasmid please refer to the pEK number) |

*Appendix 1 Continued*

| Reagent type (species) or resource | Designation | Source or reference | Identifiers | Additional information |
|---|---|---|---|---|
| Recombinant DNA reagent | Plasmid: *Pacd-3::GFP* | Kyuhyung Kim | pEK187 | Plasmid generated for generating transgenic *C. elegans* strains (for requesting plasmid please refer to the pEK number) |
| Recombinant DNA reagent | Plasmid: *pDEST Pacd-5::acd-5 cDNA::mKate2* | This study | pEK190 | Plasmid generated for generating transgenic *C. elegans* strains (for requesting plasmid please refer to the pEK number) |
| Recombinant DNA reagent | Plasmid: *Pdel-5::GFP* | Kyuhyung Kim | pEK195 | Plasmid generated for generating transgenic *C. elegans* strains (for requesting plasmid please refer to the pEK number) |
| Recombinant DNA reagent | Plasmid: *KSM acd-3 cDNA* | This study | pEK207 | Plasmid generated for generating cRNA for the expression of channels in *Xenopus oocytes* (for requesting plasmid please refer to the pEK number) |
| Recombinant DNA reagent | Plasmid: *KSM flr-1 cDNA* | This study | pEK215 | Plasmid generated for generating cRNA for the expression of channels in *Xenopus oocytes* (for requesting plasmid please refer to the pEK number) |
| Recombinant DNA reagent | Plasmid: *KSM del-5 cDNA* | This study | pEK228 | Plasmid generated for generating cRNA for the expression of channels in *Xenopus oocytes* (for requesting plasmid please refer to the pEK number) |
| Recombinant DNA reagent | Plasmid: *pDEST Pacd-5::acd-5 cDNA* | This study | pEK262 | Plasmid generated for generating transgenic *C. elegans* strains (for requesting plasmid please refer to the pEK number) |
| Recombinant DNA reagent | Plasmid: *pDEST Pacd-5::acd-5(ok2657) cDNA* | This study | pEK286 | Plasmid generated for generating transgenic *C. elegans* strains (for requesting plasmid please refer to the pEK number) |
| Recombinant DNA reagent | Plasmid: *pDEST Pges-1::acd-5(ok2657) cDNA* | This study | pEK287 | Plasmid generated for generating transgenic *C. elegans* strains (for requesting plasmid please refer to the pEK number) |
| Recombinant DNA reagent | Plasmid: *pDEST Pacd-3::acd-3 gDNA::mKate2* | This study | pEK337 | Plasmid generated for generating transgenic *C. elegans* strains (for requesting plasmid please refer to the pEK number) |
| Recombinant DNA reagent | Plasmid: *pDEST Pdel-5::Nsig::mKate2::del-5 gDNA* | This study | pEK338 | Plasmid generated for generating transgenic *C. elegans* strains (for requesting plasmid please refer to the pEK number) |
| Software, algorithm | Matlab | MathWorks | https://www.mathworks.com | |
| Software, algorithm | Fiji (ImageJ). | *Schneider et al., 2012* | https://imagej.nih.gov/ij/ | |
| Software, algorithm | NEBiocalculator | New England Biolabs Inc | https://tmcalculator.neb.com/ | |
| Software, algorithm | NEBaseChanger | New England Biolabs Inc | https://nebasechanger.neb.com | |
| Software, algorithm | Graphpad | GraphPad Software, https://www.graphpad.com | Prism 9.0.2 | |
| Software, algorithm | Robocyte2+ | Multichannel Systems | https://www.multichannelsystems.com/products/roboocyte2 | |
| Software, algorithm | Neurontracker | Ithai Rabinowitch *Rabinowitch et al., 2013*; *Rabinowitch, 2020* | https://github.com/ithairab/NeuronTracker | |
| Other | Roboocyte2 | Multichannel Systems | https://www.multichannelsystems.com/products/roboocyte2 | Fully-automated system for screening of ligand-gated channels based on the standard *Xenopus* oocyte expression system. |
| Other | Roboinject | Multichannel Systems | https://www.multichannelsystems.com/products/roboinject | Fully-automated system for injecting cRNA into *Xenopus* oocytes. |

