## [Editor Report]

In this study, Kaulich and colleagues report an intriguing interplay of different proton-sensitive ion channels at different locations in the intestine of *C. elegans*. They provide both in-depth biophysical characterisations of different channel complexes as well as their in vivo involvement in the generation of calcium waves for the rhythmic defecation motor program of the worm.

---

## [Decision Letter]

**Decision letter after peer review:**

Thank you for submitting your article "Distinct roles for two *Caenorhabditis elegans* acid-sensing ion channels in an ultradian clock" for consideration by *eLife*. Your article has been reviewed by 3 peer reviewers, one of whom is a member of our Board of Reviewing Editors, and the evaluation has been overseen by a Reviewing Editor and Piali Sengupta as the Senior Editor. The reviewers have opted to remain anonymous.

The reviewers have discussed their reviews with one another and share their enthusiasm about your work. However, their reviews highlight multiple gaps in your study that require some extensive and essential revisions. The Reviewing Editor has drafted this to help you prepare a revised submission.

Please find below a detailed list of all reviewer comments. Among these, we found most essential to:

(i) Provide complete tissue specific rescue- and subcellular localization studies of the respective channels (R1#1, R2#9-10),

(ii) Perform more detailed pH measurements in both intestine and the pseudocoelom (R2#9-10),

(iii) Address the discrepancy between the genetic effects on intestinal calcium waves and apparently normal pBoc and Exp contractions (R2#11), including an improved discussion of the differential effects you observe (R3#19),

(iv) Provide a biophysical characterization of the putative FLR-1/ACD-3/DEL-5 complex, like you nicely did for ACD-5 (R2#12, R3#18).

We believe that the remaining points can be easily addressed with standard analyses and text revisions. We are looking forward to reading a revised manuscript.

*Reviewer #1 (Recommendations for the authors):*

(1) Is expression of flr-1, acd-3 and del-5 restricted to the intestine as partially suggested by Figure 4A-B. The authors mention a previous study that acd-3 and del-5 are also expressed in neurons implicated in the defecation rhythm; can this be confirmed with the reported marker line here? And how about flr-1? I think the study could be improved by tissue specific rescue experiments of all reported phenotypes (incl. Figure 7) the of these three genes.

(2) The authors report for some of their assays "excessive" long cycles in some mutants (e.g. Figure 3D,E); these findings should explicitly quantified with the adequate comparison groups; for example using K-S test complimentary cumulative distribution or any appropriate test.

*Reviewer #2 (Recommendations for the authors):*

(1) Figure 2: The authors show that acd-5 mutants have higher pH in the anterior intestinal lumen during MAT, compared to wild type controls, as revealed by KR35 imaging. Is pH only increased in the anterior intestine during MAT, or are pH levels lower in other parts of the intestine and at other times of the cycle in acd-5 mutants? In other words it would be helpful to know whether acd-5 regulates global pH throughout the intestine or whether it only affects pH during MAT.

(2) Figure 4A: the authors conclude that the FLR-1 channels function at the basolateral surface of the intestine based on the localization shown in Figure 4A. In the images shown, it is not obvious that this is the case. They appear more diffuse throughout the intestine. Are the fusion proteins functional? Do they co-localize with known basolateral markers? Do mutants in the channel affect pseudocoelom pH? These questions should be addressed if the authors want to make conclusions regarding the subcellular site of action.

A similar analysis looking at the effect of flr-1 mutations on pseudocoelom pH would really help here (a phlourin reporter has been used for this purpose), especially since they claim that flr-1 functions on the pseudocoelom surface of the intestine, but they show no data to support this claim.

(3) The issue of why flr-1 mutants have apparently absent intestinal calcium waves, yet have near normal pBoc and Exp contractions needs to be sorted out. In Figure 6 reports a calcium wave defect, but only show some representative traces are shown, with no quantification. However, the text states that calcium oscillations are "completely disrupted". Either calcium waves aren't very important for the pBoc and Exp contractions, which does not agree with previous research or their model, or more likely, calcium waves are not "completely disrupted", but are more intact than they report. It is further unclear if the traces represent fluorescence in the posterior intestine only or of the entire intestine. Is there a defect in the anterior propagation of calcium waves? Is there a correlation in animals that do not exhibit calcium transients and a lack of pBoc, or EMC? If calcium transients are eliminated in flr-1 mutants, why do animals exhibit pBoc in 90% of cycles and EMC in 50% of cycles? Is there another oscillator, like pH in the intestine that drives these steps? Can the authors measure pH changes in the intestine during EMC to address this?

(4) Figure 4. Can the authors present an analysis of which ions pass through FLR-1/ACD-3/DEL-5, as they did for ACD-5? Knowing this, particularly if these pass calcium, may help the authors to formulate a mechanism by which FLR-1 regulates intestinal calcium dynamics.

*Reviewer #3 (Recommendations for the authors):*

(1) The authors have done a nice job in characterizing the biophysical properties of ACD-5. But their characterization of FLR-1/ACD-3/DEL-5 is a bit cursory, and the data from FLR-1/ACD-3/DEL-5 recordings is not nearly as convincing. Is the FLR-1/ACD-3/DEL-5 heteromeric channel sensitive to amiloride? The authors claim that it is a leaky cation channel, but does not provide much evidence to support this claim. For example, is it impermeable to NMDG like ACD-5 (to show it is a cation channel)? Is it a monovalent channel like ACD-5 or also permeable to divalent ions? Is it sensitive to ca^2+^ inhibition like ACD-5?

(2) The authors have proposed a nice model in Figure 8, but it is still not so clear to me why FLR-1/ACD-3/DEL-5 affects ca^2+^ wave but ACD-5 does not. Both channels are inhibited by acidic pH, though ACD-5 is also inhibited by high pH; and both channels are non-desensitizing. Their distinct subcellular localization patterns do not seem to explain it. The authors need to put up a better explanation to this interesting phenomenon in the discussion.

---

## [Author Response]

Reviewer #1 (Recommendations for the authors):(1) Is expression of flr-1, acd-3 and del-5 restricted to the intestine as partially suggested by Figure 4A-B. The authors mention a previous study that acd-3 and del-5 are also expressed in neurons implicated in the defecation rhythm; can this be confirmed with the reported marker line here? And how about flr-1? I think the study could be improved by tissue specific rescue experiments of all reported phenotypes (incl. Figure 7) the of these three genes.

Apologies for omitting the transcriptional fusion data, showing that, with our constructs, we only see expression in the intestine. We have added Figure 4—figure supplement 1 – transcriptional fusions, together with a description in the text (page 10 line 258). We have also added this information where we refer to the CENGEN data (page 12 line 331)

We agree that tissue-specific rescue is an important question. We have inserted two additional figures (Figure 5—figure supplement 3 and Table 1), showing that intestinal expression of *acd-3* or *del-5* rescues the cycle length, cycle variability, missed and weak pBocs, missed EMCs and echos phenotypes of the *acd-3/del-5* double mutant, and text describing it (page 12 line 334), as well as adding that Take-Uchi et al., (1998) previously reported that an intestinally-expressed fusion rescues *flr-1*.

(2) the authors report for some of their assays "excessive" long cycles in some mutants (e.g. Figure 3D,E); these findings should explicitly quantified with the adequate comparison groups; for example using K-S test complimentary cumulative distribution or any appropriate test.

Thank you very much for this excellent suggestion. We have added Figure 3—figure supplement 4, showing the cumulative distribution and results of a K-S test, and text referring to it (page 9 line 229).

Reviewer #2 (Recommendations for the authors):(1) Figure 2: The authors show that acd-5 mutants have higher pH in the anterior intestinal lumen during MAT, compared to wild type controls, as revealed by KR35 imaging. Is pH only increased in the anterior intestine during MAT, or are pH levels lower in other parts of the intestine and at other times of the cycle in acd-5 mutants? In other words it would be helpful to know whether acd-5 regulates global pH throughout the intestine or whether it only affects pH during MAT.

We have added a better description in the text (page 7 line 183). As shown in the Figure 2 example pictures, the posterior part of the intestine is often too low in fluorescence to quantify. The KR35 dye is fully fluorescent at pH <~2, with a pKa of 3.5, and is non-fluorescent at a pH of ~5.5. See Figure 2H, Bender et al., 2013 for a full characterization of the pH/fluorescence relationship. Thus, observing animals with low fluorescence indicates a more neutral pH during the resting phase.

(2) Figure 4A: the authors conclude that the FLR-1 channels function at the basolateral surface of the intestine based on the localization shown in Figure 4A. In the images shown, it is not obvious that this is the case. They appear more diffuse throughout the intestine. Are the fusion proteins functional? Do they co-localize with known basolateral markers? Do mutants in the channel affect pseudocoelom pH? These questions should be addressed if the authors want to make conclusions regarding the subcellular site of action.A similar analysis looking at the effect of flr-1 mutations on pseudocoelom pH would really help here (a phlourin reporter has been used for this purpose), especially since they claim that flr-1 functions on the pseudocoelom surface of the intestine, but they show no data to support this claim.

Getting better localisation data has proven problematic; where detectable, we see a diffuse distribution, as was seen for FLR-1 (in the paper from Take-uchi et al., 1998), making membrane localisation difficult to distinguish. We have tried several different fusions for each, and different concentrations, as well as CRISPR insertions (which were not detectable).

We have added additional images to figure 4A and a longer description to better emphasise that, while we can detect basolateral localisation, we cannot exclude that the channel is also elsewhere (page 10 line 265). In the discussion (page 17 line 500), we had qualified our assumption that the channel is basolateral, referring to a previous report that FLR-1 may also be localised to the luminal membrane (Take-Uchi et al., 1998). We have expanded this to also say that we cannot exclude that ACD-3 and DEL-5 are also luminal, since, in cells where expression appears to be higher, widespread diffuse localisation could prevent us from distinguishing apical localisation. We have also added another supplementary figure (Figure 4—figure supplement 5) exploring potential electrophysiological interactions of the subunits with the luminal ACD-5. We did not detect any changes in pH-sensitivity when we co-expressed any of the subunits together with ACD-5.

We have added figure 5—figure supplement 3 and table 1, showing that the fusions rescue the double mutant

We have added a new figure (figure 7), showing that flr-1 RNAi does indeed disrupt pseudocoelomic pH oscillations.

We have also added Figure 4—figure supplement 5, showing that co-expression of FLR-1, ACD-3 or DEL-5 does not alter the properties of ACD-5, which helps to support our hypothesis that these are functionally separate channels.

(3) The issue of why flr-1 mutants have apparently absent intestinal calcium waves, yet have near normal pBoc and Exp contractions needs to be sorted out. In Figure 6 reports a calcium wave defect, but only show some representative traces are shown, with no quantification. However, the text states that calcium oscillations are "completely disrupted". Either calcium waves aren't very important for the pBoc and Exp contractions, which does not agree with previous research or their model, or more likely, calcium waves are not "completely disrupted", but are more intact than they report. It is further unclear if the traces represent fluorescence in the posterior intestine only or of the entire intestine. Is there a defect in the anterior propagation of calcium waves? Is there a correlation in animals that do not exhibit calcium transients and a lack of pBoc, or EMC? If calcium transients are eliminated in flr-1 mutants, why do animals exhibit pBoc in 90% of cycles and EMC in 50% of cycles? Is there another oscillator, like pH in the intestine that drives these steps? Can the authors measure pH changes in the intestine during EMC to address this?

Apologies, we have described our observations poorly. We have inserted additional descriptions (page 12 line 343) to better explain that calcium transients are not eliminated, but their rhythmicity is severely disrupted We see a mix of traces completely lacking calcium peaks, traces in which very small, shorter interval (< 17 second) peaks are discernible, and traces where larger magnitude, but extremely noisy and variable, calcium fluctuations occur. The variability, and particularly the noisy fluctuations, make further meaningful quantification difficult. We cannot satisfactorily identify all pBoc events in these videos, particularly the weaker ones, so cannot correlate this information with the calcium transients. However, the mix of transients observed appears to correlate well with the mix of long, short and “normal” cycle intervals, weak and missed pBocs, and missed EMCs observed.

We have included 2 additional supplemental figures (Figure 6—figure supplements 1 and 2) to show all of the remaining ca^2+^ traces, illustrating this diversity, along with wild type controls.

We have also updated the legends for these figures, to make clear that we are imaging in the posterior cells only.

We have added an additional figure (figure 7), where we show that flr-1 RNAi disrupts both intracellular and pseudocoelomic pH oscillations, together with a description in the text (page 13 line 358). The dip in pseudocoelomic pH that is thought to trigger pBoc is severely attenuated, agreeing well with the pBoc phenotype. Likewise, intracellular pH exhibits a similar basal level to wild type, but the acidification at pBoc is less extensive.

(4) Figure 4. Can the authors present an analysis of which ions pass through FLR-1/ACD-3/DEL-5, as they did for ACD-5? Knowing this, particularly if these pass calcium, may help the authors to formulate a mechanism by which FLR-1 regulates intestinal calcium dynamics.

We agree that this is an important question. We have added panels (Figure 4E, F, G) to show the ion selectivity, as well as the lack of amiloride sensitivity, along with a description (page 11 line 288).

Reviewer #3 (Recommendations for the authors):(1) The authors have done a nice job in characterizing the biophysical properties of ACD-5. But their characterization of FLR-1/ACD-3/DEL-5 is a bit cursory, and the data from FLR-1/ACD-3/DEL-5 recordings is not nearly as convincing. Is the FLR-1/ACD-3/DEL-5 heteromeric channel sensitive to amiloride? The authors claim that it is a leaky cation channel, but does not provide much evidence to support this claim. For example, is it impermeable to NMDG like ACD-5 (to show it is a cation channel)? Is it a monovalent channel like ACD-5 or also permeable to divalent ions? Is it sensitive to ca^2+^ inhibition like ACD-5?

We absolutely agree that it is important to include this data. We have added panels (Figure 4E, F, G) to show the ion selectivity and lack of amiloride sensitivity, along with a description (page 11 line 288).

We have also added Figure 4 —figure supplement 5, showing that co-expression of FLR-1, ACD-3 or DEL-5 does not alter the properties of ACD-5.

(2) The authors have proposed a nice model in Figure 8, but it is still not so clear to me why FLR-1/ACD-3/DEL-5 affects ca^2+^ wave but ACD-5 does not. Both channels are inhibited by acidic pH, though ACD-5 is also inhibited by high pH; and both channels are non-desensitizing. Their distinct subcellular localization patterns do not seem to explain it. The authors need to put up a better explanation to this interesting phenomenon in the discussion.

We agree that there are questions remaining, particularly how precisely the FLR-1 channel exerts its effect on ca^2+^ and calmodulin signalling. We have added additional discussion on this topic, including that the low currents observed for the FLR-1 channel would be more consistent with a “leak” channel that is open for much of the cycle, and discussion of the likely role of spatially confined signalling events (page 58).